# Ezh2 is essential for the generation of functional yolk sac derived erythro-myeloid progenitors

Wen Hao Neo [1,2 ✉], Yiran Meng [3], Alba Rodriguez-Meira[1], Muhammad Z. H. Fadlullah[2], Christopher A. G. Booth [1], Emanuele Azzoni [3,6], Supat Thongjuea [4], Marella F. T. R. de Bruijn [3], Sten Eirik W. Jacobsen[1,5], Adam J. Mead [1,7 ✉] & Georges Lacaud [2,7 ✉]

Yolk sac (YS) hematopoiesis is critical for the survival of the embryo and a major source of tissue-resident macrophages that persist into adulthood. Yet, the transcriptional and epigenetic regulation of YS hematopoiesis remains poorly characterized. Here we report that the epigenetic regulator Ezh2 is essential for YS hematopoiesis but dispensable for subsequent aorta–gonad–mesonephros (AGM) blood development. Loss of EZH2 activity in hemogenic endothelium (HE) leads to the generation of phenotypically intact but functionally deficient erythro-myeloid progenitors (EMPs), while the generation of primitive erythroid cells is not affected. EZH2 activity is critical for the generation of functional EMPs at the onset of the endothelial-to-hematopoietic transition but subsequently dispensable. We identify a lack of Wnt signaling downregulation as the primary reason for the production of non-functional EMPs. Together, our findings demonstrate a critical and stage-specific role of Ezh2 in modulating Wnt signaling during the generation of EMPs from YS HE.

[1] Haematopoietic Stem Cell Biology Laboratory, MRC Molecular Haematology Unit, MRC Weatherall Institute of Molecular Medicine, Radcliffe Department of Medicine, University of Oxford, Oxford OX3 9DS, UK. [2] Stem Cell Biology Group, Cancer Research UK Manchester Institute, The University of Manchester, Macclesfield SK10 4TG, UK. [3] MRC Molecular Haematology Unit, MRC Weatherall Institute of Molecular Medicine, Radcliffe Department of Medicine, University of Oxford, Oxford OX3 9DS, UK. [4] MRC WIMM Centre for Computational Biology, MRC Weatherall Institute of Molecular Medicine, University of Oxford, Oxford OX3 9DS, UK. [5] Department of Medicine Huddinge, Center for Hematology and Regenerative Medicine and Department of Cell and Molecular Biology, Karolinska Institutet and Karolinska University Hospital, Stockholm, Sweden. [6] Present address: School of Medicine and Surgery, University of Milano-Bicocca, Monza, Italy. [7] These authors jointly supervised this work: Adam J. Mead, Georges Lacaud. ✉email: wenhao.neo@cruk.manchester.ac.uk; adam.mead@imm.ox.ac.uk; georges.lacaud@cruk.manchester.ac.uk

The first (primitive) wave of hematopoiesis in mouse embryonic development is mainly characterized by the generation of primitive erythroid progenitors in the yolk sac (YS) from E7.5[1,2]. Subsequently, the emergence of YS derived erythro-myeloid progenitors (EMPs) around E8.25 marks the onset of definitive hematopoiesis[1,2]. EMPs provide an essential source of erythroid and myeloid cells to sustain embryo development prior to the generation of multipotent and long-term self-renewing hematopoietic stem cells (HSCs) in the aorta–gonad–mesonephros (AGM), and their subsequent expansion in the fetal liver (FL)[3]. Defect in primitive hematopoiesis results in early lethality before E10.5[4], a lack of EMP causes lethal embryonic anemia around E13.5, while the absence of HSC leads to perinatal lethality[5–7]. Importantly, both YS-derived primitive erythroid, YS-derived EMP and AGM HSC arise from a similar transient subpopulation of endothelial cells called hemogenic endothelium (HE) through a process of endothelial-to-hematopoietic transition (EHT)[8–12]. However, to which extent the mechanisms that direct EHT in the YS versus AGM are distinct remains unknown.

Ezh2, a core component of the polycomb repressive complex 2 (PRC2), exerts its epigenetic function through mediating the trimethylation of H3K27, a repressive histone modification[13]. PRC2 plays an indispensable role during early mouse embryo development as loss of any PRC2 core component leads to lethality prior to gastrulation[14–16]. Extensive studies on various conditional knockout (cKO) models have indicated that Ezh2 is not intrinsically required for functional adult and fetal HSCs despite possible lower phenotypic HSC number[17–21]. However, the involvement of Ezh2 in YS hematopoiesis has not been directly investigated. We previously demonstrated that inactivation of EZH2 methyltransferase activity via the deletion of SET domain, mediated either by Tie2-Cre (Tie2-Ezh2-KO) or by Vav-iCre (Vav-Ezh2-KO), results in a striking difference in survival[17]. Vav-Ezh2-KO embryos, in which deletion is activated at the onset of the hematopoietic stage[22], survive until the perinatal stage[17]. In contrast, Tie2-Ezh2-KO embryos, in which deletion is activated at the endothelial stage[23], die around E13.5 from severe anemia[17]. While this severe anemia may partly involve cell-extrinsic mechanisms[17], the development of lethal embryonic anemia at E13.5 in Tie2-Ezh2-KO embryos also suggests a possible defect of YS EMP[6,7]. We, therefore, hypothesized that loss of Ezh2 function in YS endothelial cells might impact YS hematopoiesis, and in particular, the generation of EMPs.

Here we show, by combining the phenotypic analysis of Ezh2 cKO mouse models with functional assays, that although phenotypic EMPs are generated in vivo, they are functionally defective. To gain insights into the stages affected by the loss of Ezh2, we took advantage of the in vitro mouse embryonic stem cell (mESC) hematopoietic differentiation system that closely recapitulates key developmental stages of YS hematopoiesis[8,24,25]. We observed that Ezh2 is required at the onset of the EHT but quickly becomes expendable at later stages. We then identified dysregulation of Ezh2 targeted Wnt signaling as the primary reason for the generation of non-functional EMPs. Our results reveal an essential role of Ezh2 in the temporal epigenetic regulation of YS EMPs generation.

## Results

**Functional EMP generation requires Ezh2 in endothelium.** We previously demonstrated that Tie2-Ezh2-KO results in lethality around E13.5[17]. In contrast, Vav-Ezh2-KO embryos survive into late stages of gestation[17]. The early lethality in Tie2-Ezh2-KO embryos, associated with severe anemia, prompted us to investigate the potential defects in YS hematopoiesis. Quantification of primitive erythroid cells and replating assays indicated that the primitive wave of hematopoiesis was not impacted in Tie2-Ezh2-KO embryos (Supplementary Fig. 1). We next evaluated potential defects in YS EMPs, phenotypically defined as KIT$^+$CD41$^+$CD16/32$^+$ (Fig. 1a; Supplementary Fig. 2)[2]. In line with their survival until the latest stages of gestation, the number of EMPs in Vav-Ezh2-KO YS was similar to those in wildtype (WT) embryos (Fig. 1b–d). Although we observed a minor reduction in the number of phenotypic EMP in Tie2-Ezh2-KO YS, these cells expanded normally in vivo, presented a normal morphology[26], and were neither apoptotic nor senescent (Fig. 1b; Supplementary Fig. 3a–d). Furthermore, the size of RUNX1$^+$KIT$^+$ hematopoietic cell clusters, representing EMPs emergence through EHT, was not affected in Tie2-Ezh2-KO YS (Fig. 1c, d). To confirm high recombination efficiency of the Cre-lines in EMPs, we used a R26-LSL-tdTomato reporter. Both Tie2-Cre and Vav-iCre showed close to 100% deletion efficiency in E10.5 YS EMPs (Fig. 1e). We also demonstrated loss of expression of the deleted Ezh2 exon in both Tie2-Ezh2-KO and Vav-Ezh2-KO EMPs (Fig. 1f). The subtle reduction of phenotypic EMP observed in Tie2-Ezh2-KO YS is unlikely to be the leading cause of lethal embryonic anemia. Therefore, we further assessed ex vivo the colony output capability of Ezh2-inactivated phenotypic EMP. Strikingly, despite the presence of phenotypic EMPs in Tie2-Ezh2-KO YS, none of these purified E10.5 YS EMPs produced any erythroid or myeloid colonies. In contrast, purified Vav-Ezh2-KO E10.5 YS EMPs generated similar numbers and types of colonies as their WT counterparts (Fig. 1g). Live imaging indicated that Tie2-Ezh2-KO EMPs, in contrast to WT EMPs, stopped proliferating after several rounds of division (Supplementary Fig. 3e; Supplementary Movies 1 and 2). Notably, neither E9.5 nor E10.5 Tie2-Ezh2-KO YS cells were able to produce any myeloid/definitive erythroid colonies (Fig. 1h). This excludes the possibility that our observation was caused by changes in EMP surface marker expression. Together, these data indicate an essential role for Ezh2 in the generation of functional YS EMP.

**Ezh2 loss affects transcription and chromatin landscape.** To identify genome-wide molecular changes in EMPs induced by the inactivation of Ezh2 in either endothelium or hematopoietic cells, we compared by RNA-seq the global gene expression profile of Tie2-Ezh2-KO and Vav-Ezh2-KO E10.5 YS EMPs with their corresponding WT control (Tie2-Ezh2-WT, $n = 3$; Vav-Ezh2-WT, $n = 2$; Tie2-Ezh2-KO, $n = 5$; Vav-Ezh2-KO, $n = 5$; 100 cells per biological replicates). Interestingly, Tie2-Ezh2-KO EMPs clustered separately from the Vav-Ezh2-KO and WT EMPs (Fig. 2a, b). We found 613 significantly ($P < 0.05$, RPKM $> 0.2$) differentially expressed (DE) genes in Tie2-Ezh2-KO EMPs (Fig. 2a, b; Supplementary Data 1). These genes were mostly upregulated ($n = 518$, 85% of DE genes) as would be expected following the inactivation of a core member of PRC2, which mainly has a repressive role on gene expression[27]. Gene set enrichment analysis (GSEA) indicated that the expression of genes previously identified as enriched in endothelial (EC) and HE cell populations[28] were upregulated within Tie2-Ezh2-KO EMPs in comparison with WT counterparts, indicating a failure to properly downregulate endothelial cell transcriptional programs in the absence of Ezh2 (Fig. 2c). In contrast, Tie2-Ezh2-KO EMPs expressed similar levels of hematopoietic genes (previously defined as enriched in HSCs[28]) as WT EMPs (Fig. 2c). Tie2-Ezh2-KO EMPs also expressed normal levels of EMP signature genes[26] without any obvious bias towards specific lineages (Supplementary Fig. 4a, b). A much smaller number of genes was downregulated in Tie2-Ezh2-KO EMP ($n = 95$, 15% of DE genes). Unlike the Tie2-Ezh2-KO EMP, the gene expression profile of Vav-Ezh2-KO EMP was highly similar to WT EMP as indicated by differential gene expression, principal component analysis (PCA) and GSEA (Fig. 2a, b; Supplementary Fig. 4c), with only

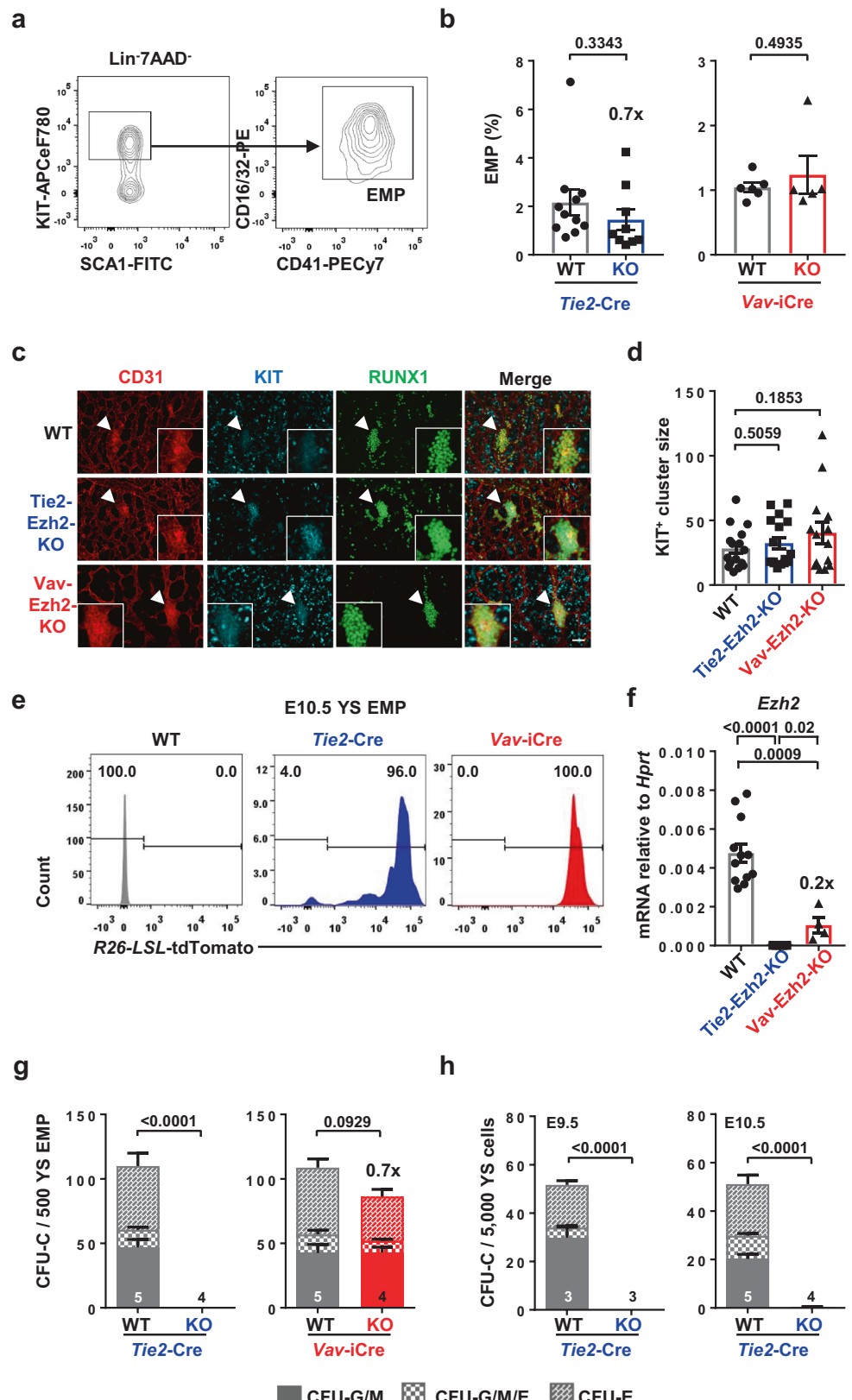

14 genes being significantly upregulated ($n = 8$) or downregulated ($n = 6$). Collectively, these results indicate that the inactivation of Ezh2 in endothelium leads to increased/sustained expression of a large number of genes in EMPs. In contrast, the deletion of Ezh2

in emerging hematopoietic cells has no significant effects on global gene expression patterns in EMPs.

To further dissect the molecular mechanisms regulated by Ezh2 in EMPs and to infer potential Ezh2 target genes, we

**Fig. 1 Ezh2 is essential for functional EMP emergence. a** Gating strategy used to define EMP (LIN⁻SCA1⁻KIT⁺CD16/48⁺CD41⁺) population within the total YS live singlet population. **b** Percentage of EMP per E10.5 YS (Tie2-Ezh2-WT, $n = 11$; Tie2-Ezh2-KO, $n = 9$; Vav-Ezh2-WT, $n = 6$; Vav-Ezh2-KO, $n = 5$; 3 independent experiments). **c** Representative whole-mount immunostaining of E10.5 YS ($n = 3$ YS, 2 independent experiments) using antibodies against CD31 (red, endothelial marker), KIT (cyan, hematopoietic marker) and RUNX1 (green). White arrowhead enlarged hematopoietic cluster. Scale bars, 50 μm. **d** Number of KIT⁺ cells per hematopoietic cluster in E10.5 YS (WT, $n = 16$; Tie2-Ezh2-KO, $n = 16$; Vav-Ezh2-KO, $n = 14$; 3 biologically independent replicates; 2 independent experiments). **e** Representative FACS histogram showing the recombination efficiency of Tie2-Cre ($n = 2$ YS) and Vav-iCre ($n = 3$ YS) in E10.5 YS EMP (2 independent experiments). **f** Ezh2 mRNA expression level in E10.5 YS EMPs (WT, $n = 12$; Tie2-Ezh2- KO, $n = 5$; Vav-Ezh2-KO, $n = 4$; 2 independent experiments). **g, h** CFU-C numbers per 500 sorted YS EMPs (**g**) or 5000 total YS cells (**h**). The numbers of biologically independent replicates are indicated at the bottom of each column (3 independent experiments). Two-tailed $t$-test was used to assess statistical significance in **b**, **d**, **f**. Two-way ANOVA was used to assess statistical significance in **g**, **h**. Error bars represent ± or + SEM.

performed H3K27me3 CUT&RUN on 500 and 15,000 WT E10.5 YS EMPs (Supplementary Data 2). We detected 5,352 peaks corresponding to 2,420 potential Ezh2 target genes with 500 EMPs (±3 kb from transcription start site (TSS); $P < 0.01$) and 7,093 peaks corresponding to 4,129 potential Ezh2 target genes with 15,000 EMPs (±3 kb from TSS; $P < 0.05$) respectively. The P value stringency was increased for the 500 EMPs dataset to correct for the inherent high background noise associated with this low cell number sample. Nearly 99% of the potential Ezh2 targets identified from the 500 EMPs were present in the 15,000 EMPs H3K27me3 CUT&RUN dataset (Supplementary Fig. 5a). As the lower input number (500 EMPs) probably limits the detection of Ezh2 targets, we considered all the targets identified with 15,000 EMPs as potential Ezh2 target genes. The majority of these Ezh2 target genes were upregulated and enriched in the Tie2-Ezh2-KO EMPs in comparison to WT EMPs (55%; Fig. 2c). A large proportion of EC and HE enriched genes were targets of Ezh2, unlike the hematopoietic genes, which is in line with their upregulation Tie2-Ezh2-KO EMPs detected by GSEA (Fig. 2d; Supplementary Data 3 and 4). This is further supported by gene ontology (GO) analysis which revealed a significant enrichment of EC-related genes (Cadherin signaling pathway and Angiogenesis) in the Ezh2 target genes (Supplementary Fig. 5b).

As Ezh2 is an epigenetic regulator, we next investigated whether the transcriptomic differences between Tie2-Ezh2-KO, Vav-Ezh2-KO, and WT EMPs are linked to differences in chromatin accessibility. We therefore performed ATAC-seq on sets of 500 EMPs for each genotype (Tie2-Ezh2-WT, $n = 3$; Vav-Ezh2-WT, $n = 3$; Tie2-Ezh2-KO, $n = 5$; Vav-Ezh2-KO, $n = 3$). In line with our RNA-seq data, inactivation of Ezh2 by Tie2-Cre mainly led to increased chromatin accessibility that was concentrated around TSS (Fig. 2e; Supplementary Fig. 6). Overall, we detected 3183 regions with increased, and 672 regions with decreased, accessibility corresponding to 2513 genes and 640 genes (±5 kb from TSS, $P < 0.05$) respectively. The significantly DE genes in Tie2-Ezh2-KO EMPs demonstrated a significant positive correlation between increased gene expression and augmented chromatin accessibility (Fig. 2f; $R^2 = 0.24$, $P < 2.2 \times 10^{-16}$). Conversely, a threshold-free comparison between differential accessibility (≤ or ≥ 1.5FC, $P < 0.05$) and RNA expression level also indicated a positive global correlation between increased chromatin accessibility and higher gene expression (Fig. 2g, $P < 0.0001$). Further supporting the results of GSEA, analysis on the TSS regions of Ezh2 target (Hoxb13), EC- (Col5a1) and HEC- (Igfbp2) genes revealed a positive association between higher chromatin accessibility and increased gene expression, while hematopoietic (Runx1, Gfi1b, Itga2b) genes were not affected (Supplementary Fig. 6). Altogether, these data establish a link between changes in chromatin accessibility and alterations of transcription in Tie2-Ezh2-KO EMPs that are associated with the generation of functionally impaired EMP upon Ezh2 deletion.

**Functional EMP generation requires Ezh2 at EHT initiation.** Our results revealed a specific requirement for Ezh2 in EHT in the YS as opposed to the AGM where previous in vivo studies have indicated that Ezh2 is dispensable[17,19]. However, as Tie2-Cre induces deletion as soon as endothelial cells develop and thereafter[23], it is not clear if Ezh2 is required for productive EHT at specific times of YS blood cell development. To address this question, we performed pharmacological inhibition of Ezh2 on ex vivo explant cultures. In agreement with our embryo data, inactivation of Ezh2 with the specific inhibitor GSK126[29] affected the hematopoietic output of WT YS explant but not AGM explant cultures (Fig. 3a; Supplementary Fig. 7a), supporting the notion that Ezh2 is mainly required for the generation of YS but not AGM derived hematopoietic cells. Importantly, the hematopoietic impact of Ezh2 inhibition was only significant with E8.5 WT YS and not later E9.5 WT YS (Fig. 3a; Supplementary Fig. 7a). As such, these results indicate a more prominent role of Ezh2 in the generation of functional EMPs than in the production of other hematopoietic precursors.

While our in vivo and ex vivo analysis provided significant insights into the requirement of Ezh2 in the generation of functional EMPs through EHT, they could not pinpoint the exact stage of EHT where Ezh2 exert its roles. We therefore turned to in vitro murine Embryonic Stem Cell (mESC) hematopoietic differentiation, that recapitulates YS hematopoietic development, and allows the detailed analysis of different stages of EHT[8,24]. In this system, mesoderm (MES) derived hemangioblast (HB) generates hematopoietic progenitors (HP) through two hemogenic endothelial stages (HE1 and then HE2). We first assessed the expression of Ezh2 and its homolog Ezh1[30] in published genome-scale resource of mESC blood cell differentiation, including ES, MES, HB, HE, HP, and macrophage (Mφ) enriched cell populations[31]. Ezh2 expression was maintained at a high expression level throughout in vitro hematopoietic development until terminal differentiation to myeloid cells. In contrast, Ezh1 only started to be expressed in differentiated Mφ (Supplementary Fig. 7b). As such, these data suggest that Ezh2 is more likely to regulate YS EHT than its homolog Ezh1, in line with the finding that Ezh1 is dispensable for maintaining fetal hematopoiesis[32]. To examine the effect of Ezh2 inhibition on the dynamics of EHT and subsequent HP formation, we differentiated HB enriched cells, isolated from day three embryonic bodies (EB), in the presence of GSK126 or the Ezh2/Ezh1 dual inhibitor UNC1999 (Supplementary Fig. 7c)[33]. Neither GSK126 nor UNC1999 treatment at day 1 and day 2 of cultures had any impact on the formation of HE (TIE2⁺KIT⁺) and HP (CD41⁺) populations evaluated at day 3 of cultures (Fig. 3b, c). Also, the EHT process appeared normal in treated cultures as indicated by the emergence of HE core-like structures and colonies of round budding cells in time-lapse imaging (Fig. 3d; Supplementary Movies 3 and 4). These results indicate that Ezh2 inhibition does not morphologically affect the EHT process or the generation of phenotypic HE and HP. However, Ezh2 inhibition significantly impaired the ability of the cells generated in these cultures to produce

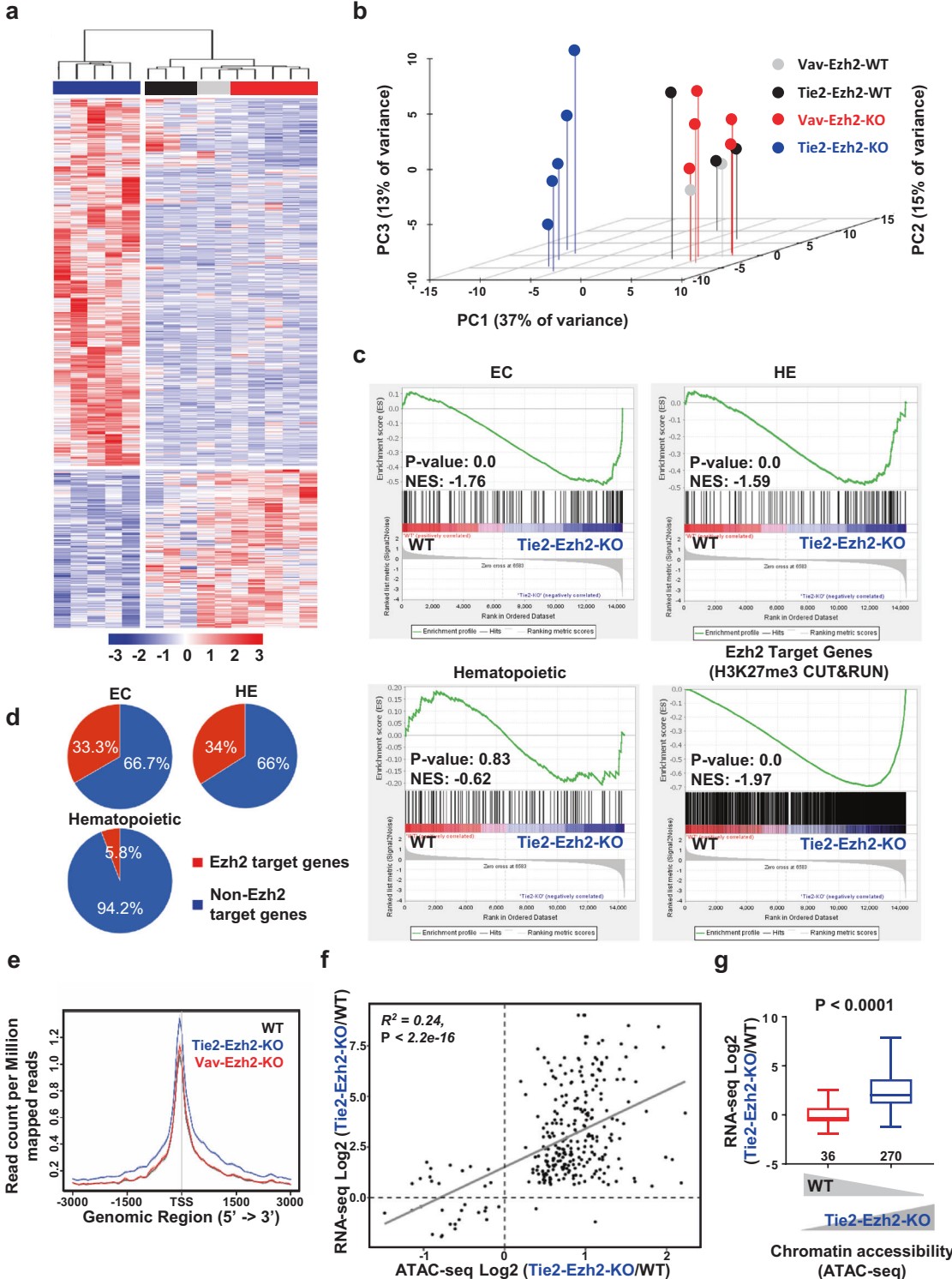

**Fig. 2 Tie2-Ezh2-KO EMPs maintain endothelial and HE gene expression signatures. a** Hierarchical clustering and heatmap showing expression levels of all significantly DE genes between Vav-Ezh2-WT ($n = 2$), Tie2-Ezh2-WT ($n = 3$), Vav-Ezh2-KO ($n = 5$) and Tie2-Ezh2-KO ($n = 5$) EMPs. Each replicate represents 100 purified EMPs from individual YS at E10.5 (3 independent experiments). **b** Principal component analysis of RNA-seq data in **a**. **c** GSEA comparing WT and Tie2-Ezh2-KO EMP for EC, HE, hematopoietic and Ezh2 target gene sets. **d** Proportion of Ezh2 target genes in EC, HE and hematopoietic gene sets. **e** Enrichment of ATAC-seq signal around transcription start site (TSS) in E10.5 YS EMPs. Vav-Ezh2-WT ($n = 3$), Tie2-Ezh2-WT ($n = 3$), Vav-Ezh2-KO ($n = 3$) and Tie2-Ezh2-KO ($n = 5$). Each replicate represents 500 purified EMPs from individual YS at E10.5 (2 independent experiments). **f** Scatterplot depicting a significant positive correlation between RNA-seq (selected DE genes, $P < 0.05$, RPKM > 0.2) and ATAC-seq ($P < 0.05$) dataset. Statistical significance was determined based on the Pearson correlation coefficients. **g** Threshold-free comparison between differential accessibility ($\leq$ or $\geq$ 1.5FC, $P < 0.05$) and RNA expression level. Medians are shown as a solid black line, box indicates the upper (75%) and lower (25%) percentile, whiskers indicate min to max. The numbers of genes are indicated at the bottom of each column. Two-tailed t-test was used to assess statistical significance. DE, differentially expressed; NES, normalized enrichment score.

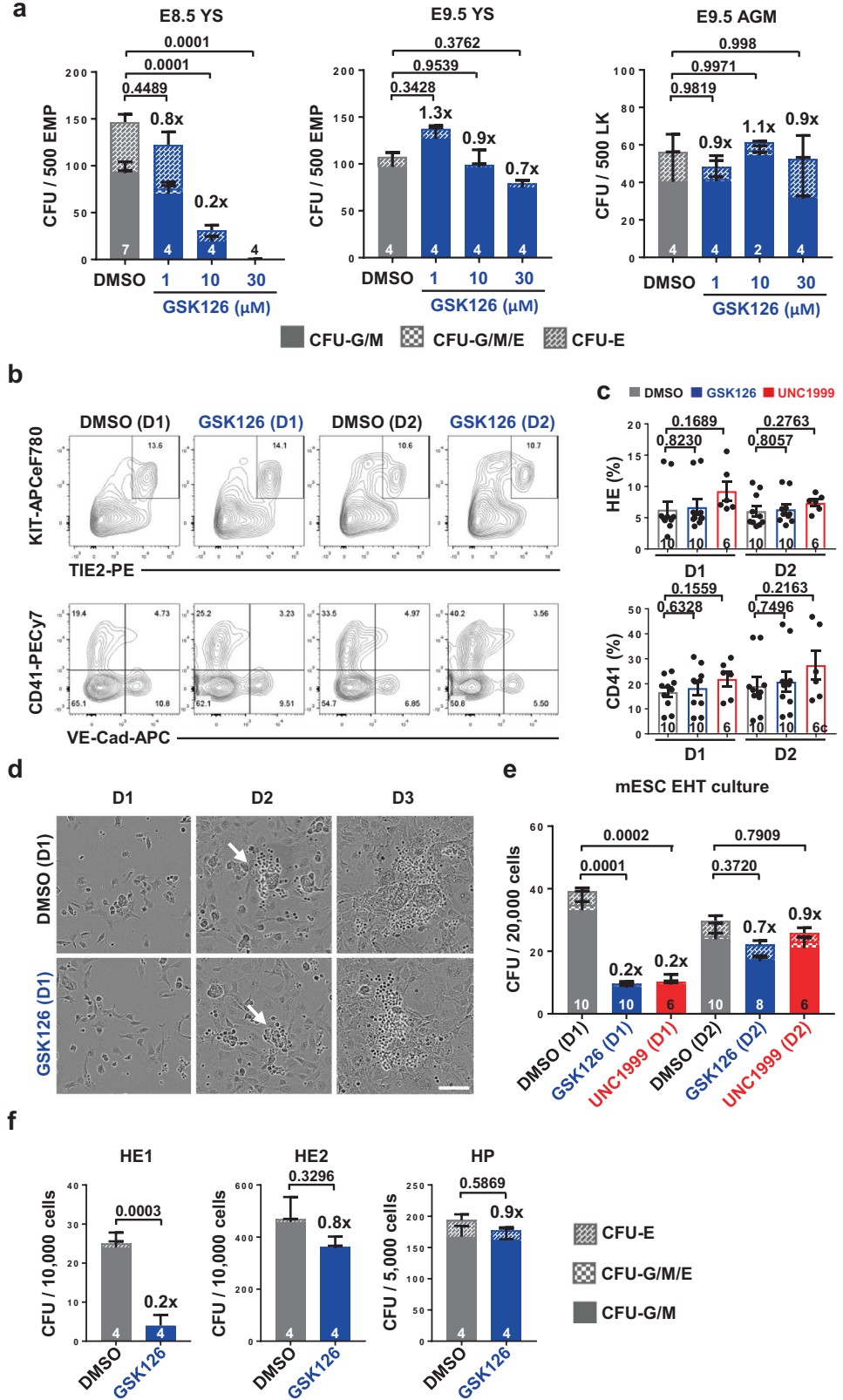

hematopoietic colonies (Fig. 3e). Importantly, delaying the Ezh2 inhibition from day 1 to day 2 of culture dramatically reduced this impact, indicating a more prominent role of Ezh2 early in the EHT. Co-inhibition of Ezh1 (UNC1999) did not exacerbate the effect (Fig. 3e). EHT is a highly dynamic non-synchronous process in bulk cultures. HE can be further separated into HE1 (TIE2$^+$KIT$^+$CD41$^{lo}$)

and HE2 (TIE2$^+$KIT$^+$CD41$^+$) based on the acquisition of CD41 expression[8]. HE2 subsequently differentiates into HP (TIE2$^-$KIT$^+$CD41$^+$) by losing TIE2 expression. We specifically inhibited Ezh2 in sorted HE1, HE2, and HP populations (Supplementary Fig. 7d). Ezh2 inhibition exhibited the most significant impact on the hematopoietic colony outputs of these cultures when added at the

**Fig. 3 Ezh2 inhibition at HE1 hampers the generation of functional hematopoietic cells. a** CFU-C numbers of E8.5 YS, E9.5 YS and AGM explant cultures treated with an increasing dosage of GSK126. Numbers of analyzed YS or AGM are indicated at the bottom of each column (2 independent experiments). **b–e** FLK1$^+$ hemangioblasts were cultured in EHT cultures with or without 1 µM GSK126 or 1 µM UNC1999 at day 1 (D1) or day 2 (D2) and analyzed at day 3 (D3) (5 independent experiments). **b** Representative FACS plot gated on singlets. **c** Percentage of HE (TIE2$^+$KIT$^+$) and hematopoietic cells (CD41$^+$), and representative images extracted from time-lapse imaging **d** showing no effect of Ezh2 inhibition on mESC hematopoietic differentiation. Arrows depict hematopoietic progenitors. Scale bars, 100 µm. **e** Measurement of the hematopoietic output of DMSO, GSK126 or UNC1999 treated D3 cultures with CFU assay. **f** CFU-C assay of DMSO or GSK126 treated HE1, HE2 and HP (4 independent experiments). Gating strategy for HE1, HE2 and HP is detailed in Supplementary Fig. 7d. The numbers of biological replicates are indicated at the bottom of each column. Two-way ANOVA was used to assess statistical significance in **a**, **e**, **f**. Two-tailed $t$-test was used to assess statistical significance in **c**. Error bars represent ± or + SEM.

HE1 stage (Fig. 3f). Taken together, our in vitro data demonstrate a specific requirement of Ezh2 for the generation of functional hematopoietic cells at the earliest HE1 stage of EHT.

**Temporal regulation of Wnt signaling by Ezh2 during EHT.** To identify the molecular pathways responsible for the functional deficiency of EMPs in the absence of Ezh2, we performed unsupervised GSEA on E10.5 YS EMP RNA-seq data with the molecular signatures database hallmark gene sets (Supplementary Data 3). Wnt β-catenin signaling was one of the most significantly upregulated pathways in Tie2-Ezh2-KO EMP (Fig. 4a). Further supporting a link between Wnt signaling and Ezh2, nearly half of Wnt signaling genes were present in our Ezh2 target gene list (Fig. 4b; Supplementary Data 4) and conversely the GO analysis on the Ezh2 target genes revealed a significant enrichment of Wnt signaling pathway genes (Supplementary Fig. 5b). Accordingly, increased chromatin accessibility was observed at the TSS of the Wnt associated genes *Gnai1*, *Shisa3*, and *Sulf2*, and their expression was upregulated specifically in Tie2-Ezh2-KO EMPs (Supplementary Fig. 6). The increase in Wnt signaling in Tie2-Ezh2-KO EMPs was confirmed by nuclear β-CATENIN staining (Fig. 4c). To experimentally evaluate the extent to which sustained Wnt signaling affects EMPs functionality, we treated mESCs EHT cultures with the Wnt agonist CHIR99021[34] (Supplementary Fig. 8a). We observed that similarly to GSK126, cultures treated at day 1 with CHIR99021 generated similar frequencies of HE and CD41$^+$ cells as their WT counterparts, but that their hematopoietic colony output was significantly decreased (Fig. 4d, e). These results indicate that activation/maintenance of the Wnt signaling pathway has a similar hematopoietic impact as Ezh2 inhibition on the EHT. Interestingly, co-treatment with GSK126 and CHIR99021 did not result in a more substantial decrease in colony output, suggesting that the two inhibitors affect a similar process (Fig. 4e). To further assess this finding, we inhibited Wnt signaling with the Wnt antagonist IWP2[35] in cultures treated with GSK126. We observed a partial rescue of the functional defect caused by Ezh2 inhibition (Fig. 4e). Also, Wnt inhibition was able to partially revert the hematopoietic impact of GSK126 Ezh2 inhibition on HE1 (Fig. 4f; Supplementary Fig. 8b). To examine if similar effects are observed with EMPs, we assessed the effect of Wnt inhibitor treatment on WT and Tie2-Ezh2-KO E10.5 YS EMPs (Supplementary Fig. 8c). In keeping with our mESC result, Wnt signaling inhibition with IWP2 alleviated, to a large extent, the hematopoietic impact of Ezh2 deletion (Fig. 4g). To further strengthen the link between Wnt signaling and EMP functionality, we treated the EMPs with another class of Wnt inhibitor (DKK1) that sequesters the LRP6 co-receptor to prevent activation of the Wnt signaling pathway upon ligand binding[36,37]. This is different from IWP2 which blocks the Wnt signaling by preventing the secretion of WNT ligands[38]. A similar rescue effect of replating efficiency of Tie2-Ezh2-KO E10.5 YS EMPs was observed with both IWP2 and DKK1 treatments, further supporting the importance of regulated

Wnt activity in EMP functionality and ruling out potential off-target effects. Noteworthily, the response to IWP2 treatment on EMPs suggests that EMPs express and secrete WNT ligands (Fig. 4g). Indeed, this is supported by our RNA-seq data which show *Wnt4* and *Wnt5b* ligands expression alongside increased *Porcn* expression in Tie2-Ezh2-KO EMPs (Supplementary Fig. 9). Importantly, all these three genes are potential Ezh2 targets (Supplementary Fig. 9). Together, this suggests that Ezh2 regulates Wnt signaling in part by repressing PORCN expression which is important to process WNT4 and WNT5b ligands for their secretion. In conclusion, our data indicate that Wnt signaling is dysregulated upon the deletion of Ezh2 and that its sustained activation is, at least partly, responsible for the production of non-functional EMP.

**Discussion**

During ontogeny, YS and AGM are the major sites of the emergence, through EHT, of EMP and HSC respectively. Due to the tremendous clinical potential of HSC in numerous therapies, there has been until recently a much stronger emphasis on investigating and understanding AGM EHT. However, there is an increasing recognition that studying YS EHT would also generate a wealth of critical information[3]. First, a complete understanding of the EHT that gives rise to HSC is achievable only if we identify how AGM EHT differs from EHT in the YS. Secondly, there is increasing evidence that EMPs derived cells are not only sustaining early embryonic development but are also a significant source of hematopoietic cells in adults such as tissue-resident macrophages (Mφ) or subtype of NK cells[3]. Additionally, tumor-associated Mφ, which modulate the tumor microenvironment and contribute to therapy failure, have been proposed to originate from EMPs[39]. Therefore, there is a clear scientific and translational interest to now further investigate the process of YS EHT and, in particular, the generation of EMPs. In this study, we established a critical and specific role of Ezh2 in modulating Wnt signaling during the generation of EMPs from YS HE.

We have previously identified a cell-extrinsic role of Ezh2 in the regulation of FL erythropoiesis, via MMP9 expression, and the intrinsic dispensability of Ezh2 for the emergence of fetal HSC in the AGM[17]. Nevertheless, the development of lethal anemia at E13.5 in Tie2-Ezh2-KO embryos is also strikingly similar to the phenotype associated with loss of EMPs, suggesting a possible defect of YS hematopoiesis which has not been previously investigated. We, therefore, hypothesized here that loss of Ezh2 function in YS endothelial cells might impact YS hematopoiesis, and in particular, the generation of EMPs. To our surprise, we only observed a minor reduction in phenotypic EMP numbers in Tie2-Ezh2-KO YS compared to Vav-Ezh2-KO YS or WT YS. However, when we further assessed ex vivo the colony output capability of Ezh2-inactivated phenotypic EMPs, they were unable to produce any erythroid or myeloid colonies. These results indicate not only a profound functional defect in EMPs generated in Tie2-Ezh2-KO embryos but also highlight the

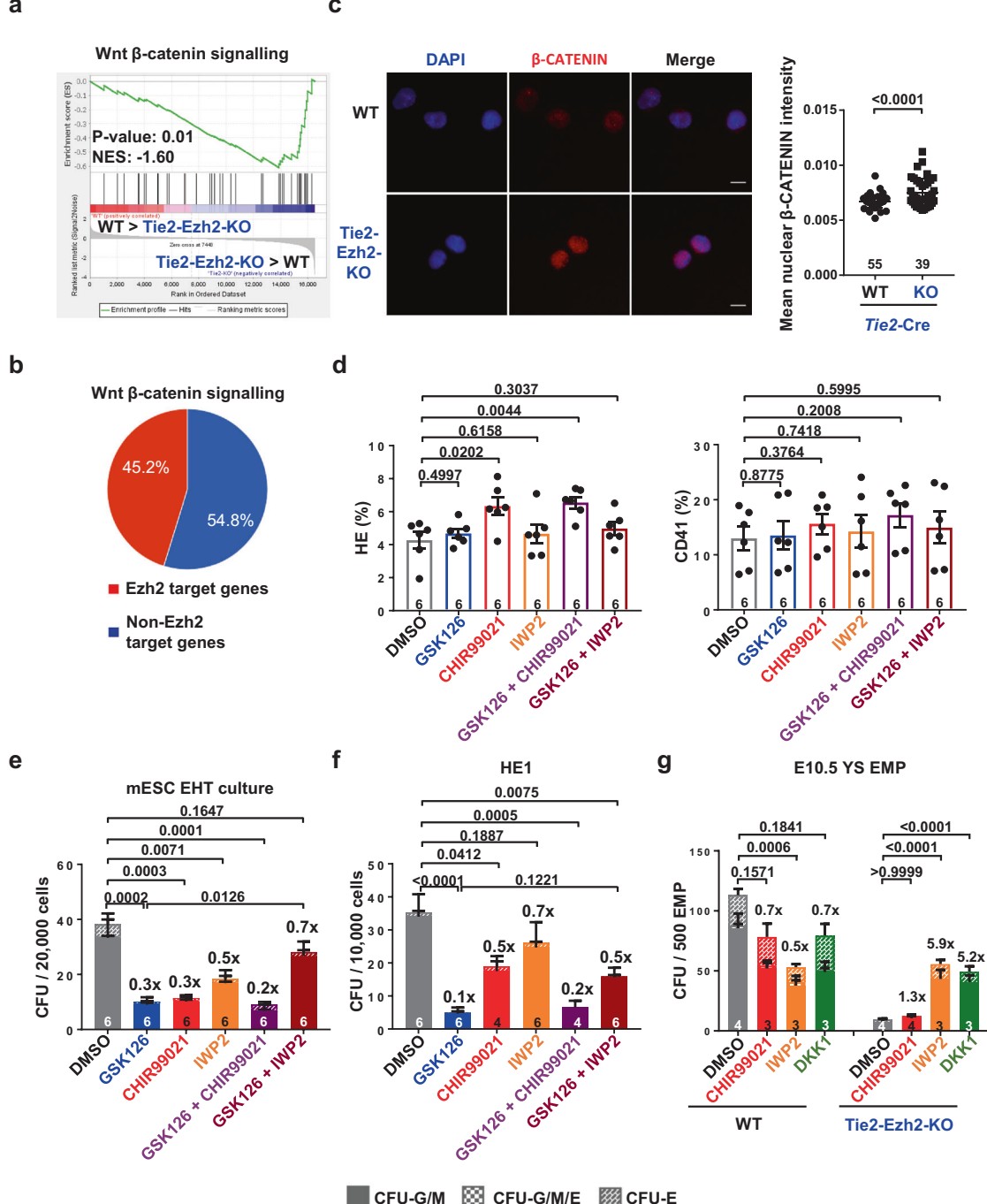

**Fig. 4 Wnt signaling inhibition rescues the functional block of hematopoiesis created by inhibiting Ezh2. a** GSEA comparing WT and Tie2-Ezh2-KO EMP for Wnt β-catenin signaling gene sets. **b** Proportion of Ezh2 target genes in Wnt signaling gene sets. **c** Representative immunostaining of E10.5 YS EMPs (2 independent experiments) using antibody against β-CATENIN (red). Nuclear is marked by DAPI staining (blue). Scale bars, 20 μm. Mean nuclear β-CATENIN intensity was analyzed. Numbers of analyzed EMPs are indicated at the bottom of the graph. **d**, **e** FLK1[+] hemangioblasts were cultured in EHT cultures with or without 1 μM GSK126 at day 1 and 3 μM CHIR99021 or 3 μM IWP2 at day 2 (3 independent experiments). **d** Percentage of HE (TIE2[+]KIT[+]) and hematopoietic cells (CD41[+]) by FACS analysis at day 3. **e** Measurement of the hematopoietic output of DMSO, GSK126, CHIR99021 or IWP2 treated D3 cultures with CFU-C assay. **f** CFU-C assay of DMSO, GSK126, CHIR99021 or IWP2 treated HE1 (4 independent experiments). The numbers of biological replicates are indicated at the bottom of each column in **e**, **f**. **g** Explant culture of DMSO, 3 μM CHIR99021, 3 μM IWP2 or 200 ng/ml DKK1 treated E10.5 YS EMPs (2 independent experiments). Numbers of analyzed YS are indicated at the bottom of each column. Two-tailed t-test was used to assess statistical significance in **c**, **d**. Two-way ANOVA was used to assess statistical significance in **e**, **f**, **g**. Error bars represent ± or + SEM.

importance to associate functional assays with phenotypic or morphologic quantifications.

The PRC2 is composed of the four major components EZH2, EED, SUZ12, and RBBP4/7[40]. The catalytic subunits EZH1 and EZH2 are mutually exclusive in the PRC2 complex, displaying in general opposite expression patterns. The fact that AGM HE is capable of generating functional HSCs without Ezh2 suggests a potential compensatory role of its homolog, Ezh1, as previously

shown in pluripotent mESCs[30]. However, this is unlikely as firstly, Ezh1 expression is much higher in adult HSCs than in FL LSK cells. Secondly, its deletion does not affect fetal hematopoiesis in contrast to a marked loss of HSCs by senescence in the adult bone marrow[32]. Finally, Ezh1 association with repression instead of promotion of multipotency[41] would be at odds with a potential role in the generation of HSCs.

The critical involvement of repressive complexes in the EHT is not unexpected, given the dramatic switch in transcriptional programs undergone by these cells. Our results indicate that Ezh2 is particularly required for YS EHT and not AGM EHT. As such, we identified the first element that is essential for YS, but dispensable for AGM hematopoiesis. In contrast, Notch signaling, vascular circulation and the transcription factor Hlf were previously shown to be critical for AGM hematopoiesis, but dispensable for YS blood development[42–46]. We previously identified lysine-specific demethylase 1 (LSD1)[47,48] as another repressive epigenetic factor implicated in EHT. Indeed, LSD1 is recruited by GFI1, a transcriptional repressor downstream of RUNX1, to epigenetically silence the endothelial program during AGM and YS EHT[49]. Although, no functional HSCs are generated in AGM upon Gfi1/Gfi1b deletion, YS cells were able to produce hematopoietic colonies in vitro despite endothelial program retention[50], further highlighting mechanistic differences between YS and AGM EHT. Although both EMPs generated in Tie2-Ezh2-KO and Gfi1/Gfi1b KO retain their endothelial signatures, only the Gfi1/Gfi1b KO EMPs are functional in vitro[50]. This finding suggests that loss of Ezh2 affects additional essential processes apart from silencing the endothelial program. The histones deacetylases HDAC1 and HDAC2 are other epigenetic repressors that are individually required but not essential for hematopoietic emergence via modulation of the TGFβ signaling pathway[51]. The deletion of both genes results in apoptosis of cells undergoing EHT and a complete lack of functional hematopoietic cell generation. Altogether, these studies have revealed that epigenetic silencers have critical and often distinct roles during either YS or AGM EHT and in particular impact on signaling pathways.

Indeed, EHT is a highly intricate process that is orchestrated by complex spatiotemporal activations and repression of signaling pathways such as fibroblast growth factor, Kit, Notch, BMP, TGFβ, retinoic acids and Wnt[52–54]. Among them, Wnt is crucial at multiple stages of mouse developmental hematopoiesis. Wnt has been shown to be critical for both EMP and HSC production as β-catenin deletion by *Cdh5*-Cre reduces EMP and HSC numbers significantly[55,56]. Interestingly, Wnt requirement was only transient as its inactivation from the hematopoietic stage by *Vav*-iCre has no impact on the HSC[55]. In contrast, Wnt need to be downregulated upon HSC emergence, and its constant activation will impair HSC formation[57]. Our data indicate that the EHT giving rise to EMPs is also sensitive to Wnt dosage as we observed both over-activation and inhibition of Wnt have a negative effect on EHT. Therefore, our study demonstrates that the Wnt dependency is conserved between YS and AGM EHT and that Ezh2 is the direct upstream epigenetic regulator of Wnt in YS EHT. This is further substantiated by a recent finding that established the regulatory role of another member of the PRC2 complex, Mtf2, on Wnt signaling during definitive hematopoiesis[58].

In summary, the current work has revealed a pronounced impact of Ezh2 inactivation on YS EHT, leading to the generation of non-functional EMPs. Loss of Ezh2 in HE results in sustained Wnt activity during YS EHT and consequently the maintenance of the endothelial signature in EMPs. These findings provide evidence of a previously unrecognized stage-specific role of Ezh2 in YS EMP emergence and demonstrate the importance of

epigenetic reprogramming during EHT. These studies highlight the usefulness of combining mouse genetic modeling, mESC differentiation system and pharmacological drug treatment in understanding the initiation of definitive hematopoiesis in the YS.

## Methods

**Mouse lines.** *Ezh2[fl/fl59]*, *Tie2-Cre[23]*, *Vav-iCre[22]*, and *Rosa26-LSL-tdTomato[60]* reporter mice have been previously described. All mice were either on C57BL/6 (CD45.2) background or backcrossed to C57BL/6 (>5 generations). All mice were bred and maintained in accordance with UK Home Office project license 30/3103 and 70/8472. All experiments were approved by the Oxford University Clinical Medicine Ethical Review Committee and CRUK-Manchester Institute Animal Welfare and Ethical Review Body. Embryonic development was estimated considering the day of vaginal plug formation as 0.5 d post- coitum (dpc) and somite pairs (sp). Both male and female mice were used. Genotyping primer sequences are provided in Supplementary Table 1.

**FACS analysis and sorting.** FACS experiments were performed using LSRII, LSRFortessa, LSRFortessa X20, Influx, AriaII, AriaIII and Aria Fusion cytometers (BD Biosciences). Data was analyzed using BD FACSDiva 8.0.1 and FlowJo software (TreeStar). Cells were incubated in Fc-block before being stained with antibodies at predetermined (titrated) optimized concentrations. Gates were set using a combination of fluorescence-minus-one controls and also populations that are known to be negative for the antigen. Clones and suppliers of all antibodies used are provided in the Supplementary Table 2.

**Fetal liver and YS tissue preparation.** Dissected fetal liver (FL) and YS, at indicated embryonic stages, were finely chopped and the obtained fragments were digested in a mix of Collagenases IV (2 mgml$^{-1}$, Worthington) and DNase I (200 Uml$^{-1}$, Calbiochem) at 37 °C with gentle agitation for 15 min. The dissociated cells were centrifuged at 300 g for 6 min and resuspended in PBS supplemented with 10% FBS.

**Whole-mount immunofluorescence staining and analysis.** YS was isolated from E10.5 embryos and fixed for 1 h in 4% paraformaldehyde/PBS on ice. Embryos were dehydrated in an increasing methanol/PBS gradient (50% methanol/PBS; then 100% methanol for 10 min at 4 °C) and stored at −20 °C in 100% methanol. Following rehydration, the samples were incubated in PBT (PBS plus 0.4% Triton X-100) with 0.2% BSA and 2% normal serum (goat, donkey, rabbit or rat) for at least 2 h at 4 °C. The samples were incubated overnight with primary antibodies (or isotype controls) diluted in PBT with BSA and normal serum and washed three times in PBT the next day (at least 1 h per wash). The samples were incubated overnight with secondary antibodies plus Hoechst (1:1000) diluted in PBT and washed three times in PBT the next day. The samples were cleared in 50% glycerol/ PBS solution for overnight at 4 °C. The samples were flat-mounted on a Superfrost slide (Thermo Scientific) with a drop of 50% glycerol/PBS solution. A coverslip was placed on top to cover the sample and secured with clear nail polish. Images were acquired with a Zeiss 780 confocal upright microscope and analyzed using Fiji and Imaris software. Clones and suppliers of all antibodies used are detailed in the Supplementary Table 3.

**ESC culture and CFU-C assay.** ESC cultures, maintenance, and differentiation and composition of the different media were described previously[24,61]. For drugs treatment, 1 µM (GSK126 and UNC1999) or 3 µM (CHIR99021 or IWP2) of the compounds were added to the EHT culture at indicated stages. CFU-C assay was performed as described previously[24]. Briefly, CFU assay was performed by culturing cells in a semi-solid methylcellulose matrix with appropriate cytokines and supplements (M3434, Stem Cell Technologies). Discrete cell clusters or colonies were counted after 7-10 days of culture. Types of colony were defined according to their distinct morphology. CFU-G/M, Colony-forming unit-granulocyte/macrophage; CFU- G/M/E, Colony-forming unit-granulocyte/macrophage/erythroid; CFU-E, Colony-forming unit-erythroid. Drugs used are listed in the Supplementary Table 4.

**Explant culture.** Whole AGM or YS were cultured on top of a 0.65 µm Durapore filter (Millipore, cat. no. DVPP02500) supported by a steel mesh in IMDM media (Gibco) supplemented with 20% fetal calf serum (Gibco), 1% penicillin/streptomycin (Gibco), 1% L-glutamine (Gibco), 0.1 mM 2-mercaptoethanol, 100 ngml$^{-1}$ IL- 3, 100 ngml$^{-1}$ SCF and 100 ngml$^{-1}$ FLT3L (all cytokines from Peprotech) at 37 °C, 5% CO$_2$ and a gas–liquid interface for 3 days. For drugs treatment, tissues were pretreated with 10 µM GSK126 for 2 h in complete media before explant culture. 3 µM IWP2 was added to the culture after 24 h of culture. Tissues were harvested after 72 h of culture for FACS analysis and YS EMP or AGM LK were sorted for CFU-C analysis. Drugs used are listed in the Supplementary Table 4.

**Time-lapse imaging.** ESC lines were cultured and differentiated as described above. For time-lapse imaging, 50,000 day 3 FLK1$^+$ cells or hundreds of sorted

E10.5 YS EMPs were plated in 24 or 96-well plates. Time-lapse imaging were acquired using an IncuCyte Zoom device (Essen Instruments).

**Immunofluorescence staining**. For β-CATENIN staining, sorted EMPs were cytospin onto microscope slides and fixed with 2% paraformaldehyde/PBS for 10 min, permeabilized with 0.2% Triton X-100 for 5 min and stained with anti-β-CATENIN antibody (Cell Signaling Technologies) overnight. Stained cells were photographed with an EVOS Imaging System (Thermofisher). Clones and suppliers of all antibodies used are provided in the Supplementary Table 3.

**Apoptosis staining**. Apoptosis (A35110, Invitrogen) staining were performed according to the manufacturer's instructions.

**RNA-seq**. 100 EMPs were prepared for RNA-seq using the SMARTer Ultra Low RNA kit for Illumina Sequencing (Clontech). cDNA libraries were pre-amplified using 16 cycles of PCR. cDNA libraries were prepared using Nextera XT (Illumina). Samples were sequenced using the Illumina NextSeq 500/550 platform, generating 75 bp single-end reads.

**RNA-seq analysis**. We used TopHat (v2.0.13)[62] to align short reads to the mouse (GRCm38/mm10) reference genome. The mapping parameter "−g 1" was used to allow one alignment to the reference for a given read. The 'featureCounts'[63] software was used to count reads on the basis of the RefSeq gene model. Reads per kilobase of transcript per million mapped reads (RPKM) values were calculated using a script in R. We normalized the RPKM values into log2 scale. PCA was performed using the DESeq2 package (1.32.0)[64] using the 'plotPCA' function. Heatmap was plotted using the pheatmap package. DESeq2 was used for differentially expressed gene analysis base on the read count. P values are derived from Wald test.

**Gene set enrichment analysis**. Gene set enrichment analysis (GSEA) was performed using GSEA software (v2.2.0) (RRID:SCR_003199). P values were estimated by permuting the genes, with the result that genes are randomly assigned to the sets while maintaining their size. The Hallmark gene sets were obtained from MSigDB (RRID:SCR_003199) and converted to mouse gene identifiers using the Biomart (RRID:SCR_006442) R package prior to analysis. EC, HE and hematopoietic gene sets were derived from the previously published gene list[28] and converted to mouse gene identifiers using Biomart. The Ezh2 target gene set was defined based on H3K27me3 CUT&RUN data. Genes with expression levels lower than 0.2 RPKM were excluded from the analysis.

**CUT&RUN**. 500 or 15,000 EMPs were prepared for CUT&RUN using the CUT&RUN kit from Cell Signaling Technologies with Rabbit (DA1E) mAb IgG XP® isotype control (Cell Signaling Technologies) and anti-H3K27me3 (07-449, Merck MilliPore). Sequencing libraries were prepared using the NEBNext® UltraTM II DNA Library Prep Kit for Illumina® (New England Biolabs) with 17 (500 EMPs) or 16 (15,000 EMPs) cycles of amplification. Libraries were sequenced using the Illumina NextSeq 500 (500 EMPs) and NovaSeq 6000 (15,000 EMPs) platform, generating 76 bp (500 EMPs) and 59 bp (15,000 EMPs) paired-end reads. Clones and suppliers of all antibodies used are listed in the Supplementary Table 5.

**CUT&RUN analysis**. Raw sequencing reads were aligned to mouse genome (mm10) using Bowtie2, version 2.2.1[65]. Peak calling was performed on the aligned BAM files using MACS2, version 2.1.2[66] using the parameter -- broad and –P value 0.05 (for the 15,000 EMPs) or 0.01 (for the 500 EMPs). P values were calculated by MACS2. Regions called as peaks were annotated in R using the Bioconductor package ChIPpeakAnno, version 3.20.1[67]. BigWig files were generated from the BAM file using the deeptools program, version 3.5.1[68].

**ATAC-seq**. 500 EMPs were sorted into a 0.5-ml DNA LoBind tube (Eppendorf #022431005) containing 23.75 μl of transposition mix (8.25 μl PBS, 2.5 μl water, 12.5 μl 2× TD Buffer, 0.25 μl 1% digitonin and 0.25 μl 10% Tween-20). 1.25 μl Tn5 transposase was added to the tube and incubated at 37 °C for 30 min in a thermomixer with shaking at 300 r.p.m. Reactions were cleaned up with Qiagen MinElute Kit. The transposed DNA was pre-amplified/indexed with 14 cycles of PCR with NEBNext High-Fidelity 2x PCR master mix. Samples were sequenced using the Illumina NextSeq 500/550 platform.

**ATAC-seq analysis**. Sequencing reads from ATAC-Seq samples are quality checked using FASTQC. All bases with a Phred quality score ≤ 20 and any adapter sequences present in the data are removed using FASTX-Toolkit. The cleaned and trimmed reads from previous steps are mapped to the mm10 reference assembly using Bowtie2 v2.2.1[65]. The resulting alignment SAM files are processed using SAMtools. v1.3.1 to remove unmapped reads, multimapping reads and mitochondrial reads. Picard.v.1.96's MarkDuplicates program is fused to filter out duplicate reads. We finally retain uniquely mapped reads that are mapped in a proper pair with a mapping quality ~20 and the SAM files are converted into BAM

files and indexed. The alignment results (BAM files) are used in MACS2 v2.1.2[66] (--shift 100 --extsize 200 --qvalue 0.01 --keep-dup 1 --slocal 10000 --SPMR --bdg) for peak calling to identify genome-wide open chromatin regions which are characterized by enriched signals (reads). Homer v4.10[69] is used to annotate the significant peaks. Differential chromatin modification analysis between wild-type and knockout sample are performed by diffReps v1.55.4[70] with negative binomial statistical tests.

**Quantitative real time PCR**. Quantitative real time polymerase chain reaction (PCR) analysis was performed using the TaqMan gene expression assays (Life Technologies). CellsDirect One-Step qRT-PCR kit (Invitrogen) was used for cDNA synthesis and pre-amplification of target genes. Cells were FACS-sorted directly into 10 μL (bulk; 50 cells) of lysis/pre-amplification buffer. 10 μL of this buffer consisted of: 2.5 μL Taqman assay mix containing all assays at 0.2x dilution, 5 μL CellsDirect 2x Reaction mix (Invitrogen), 1.2 μL CellsDirect RT/Taq mix (Invitrogen), 1.2 μL TE buffer and 0.1 μL SUPERase-In RNase Inhibitor (Ambion). Reverse transcription and target pre-amplification were performed as follows: 50 °C for 15 min; 95 °C for 2 min; 22 cycles of 95 °C for 15 s and 60 °C for 4 min. Pre-amplified product was diluted 1:5 in TE buffer and analyzed using the ABI 7500 Fast real-time PCR instrument with specific Taqman probe (listed in Supplementary Table 6) that targeting *Ezh2* deleted exon. Data were normalized to *Hprt*.

**Statistics**. All statistical analyses were performed using GraphPad Prism 7.0. All results are presented in graphs are the mean ± SEM. Each exact n value is indicated in the corresponding figure legend. No statistical method was used to predetermine sample size. No samples or animals were excluded from the analyses, and the animals were not randomized. The investigators were not blinded to allocation during experiments and outcome assessment. A two-tailed unpaired t-test was used for all analysis except CFU assays which were analyzed with two-way ANOVA. FACS and immunofluorescence data presented are representative of at least three independent experiments that yielded similar results, unless otherwise noted in the figure legends.

**Reporting summary**. Further information on research design is available in the Nature Research Reporting Summary linked to this article.

## Data availability

The RNA-seq, ATAC-seq and CUT&RUN data generated in this study have been deposited in the GEO database under accession code GSE181873. Source data are provided with this paper.

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

## Acknowledgements

We thank the Biomedical Services at University of Oxford and the Biological Resources Unit at University of Manchester for expert animal support. The authors acknowledge the contributions of the WIMM Flow Cytometry Facility, supported by the MRC HIU, MRC MHU (MC_UU_12009), NIHR Oxford BRC (131/030), John Fell Fund (101/517), EPA fund (CF182 and CF170), WIMM Strategic Alliance awards G0902418 and MC_UU_12025. This work was also supported by the MRC and Wolfson Foundation (Grant 18272) funded Wolfson Imaging Centre Oxford. We thank the staff at the Flow Cytometry, Visualisation, Irradiation & Analysis, Computational Biology Support Team and Molecular Biology Core Facility of CRUK Manchester Institute for technical support. E.A. and M.F.T.R.dB. were supported by a program in the MRC Molecular Hematology Unit Core award (MC_UU_12009/2 to M.F.T.R.dB.). This work was funded by Kay Kendall Leukaemia Fund Project Grant (KKL811), a MRC Senior Clinical Fellowship (MR/L006340/1), CRUK Senior Cancer Research Fellowship (C42639/A26988) and MRC

Molecular Haematology Unit core award (MC_UU_12009/5) awarded to A.J.M. The work in G.L. laboratory is supported by Blood Cancer UK (19014) and Cancer Research UK Manchester Institute Core Grant (C5759/A27412).

## Author contributions

W.H.N., A.J.M. and G.L. conceived, designed and supervised the project. W.H.N. performed all experiments. Y.M. performed ATAC-seq. A.R.-M. performed RNA-seq. M.Z.H.F. and S.T. performed bioinformatics analysis. C.A.G.B. generated embryos for whole-mount immunofluorescence staining. E.A. performed whole-mount immunofluorescence staining. M.F.T.R.dB., S.E.W.J., A.J.M. and G.L. provided protocol and insightful comments. A.J.M. and G.L. acquired funding. W.H.N. and G.L. wrote the manuscript. All the authors have read and approved the final manuscript.

## Competing interests

The authors declare no competing interests.
