## [Peer Review File · Nature Communications]

Ezh2 is essential for the generation of functional yolk sac derived erythro-myeloid progenitorsReviewers' Comments:

Reviewer #1:

Remarks to the Author:

This paper by Neo et al. is very interesting and relevant in that the authors clarified cell-intrinsic mechanisms that cause lethal anemia in Tie2-Ezh2-KO embryos, which has not been well understood despite that the severe anemia in Tie2-Ezh2-KO embryos was reported almost 10 years ago. The authors previously reported the cell-extrinsic role of Ezh2 in fetal liver hematopoiesis. In this paper, the authors examined the function of erythro-myeloid progenitors (EMP) in Tie2-Ezh2-KO YS. Surprisingly, they found complete diminishment of colony-forming ability in the YS, despite the presence of a normal number of phenotypic EMPs. This result suggested that EMPs were generated but not functional. Organ culture and using Ezh2 inhibitor, the authors demonstrated Ezh2 deletion did not alter endothelial-hematopoietic transition, but the hematopoietic colony-forming function. RNA- and ATAC-seq analysis identified the retention of EC features in Ezh2 deleted EMP and the upregulation of Wnt signaling. Finally, the authors tested the effect of upregulation of Wnt signaling in HECs using mouse ESC differentiation system and showed impairment of colony-forming ability when Wnt agonist was added to the HEC stage. Thus, this paper detailed the cause of impairment of EMPs in Tie2-Ezh2-KO embryos, which will be of great interest to the readers in the field of developmental hematopoiesis and epigenetics.

There are a few questions.

1. The authors claimed that EHT is not altered, and phenotypical EMPs are generated, but they are not functional. How about the morphology of EMPs? I wonder if they are real blood cells or not. Do they look like EMPs when they are stained with Giemsa staining in cytospin preparation?
2. Will EMPs from Tie2-Ezh2-KO embryos undergo apoptosis in vitro culture compared to WT?
3. Are there any defects in primitive erythropoiesis at E7.5?
4. In Fig. S3d FACS plots, the gating of HE1 looks CD41 dim, not negative. Please show FMO to show HE1 is CD41 negative.

Reviewer #2:

Remarks to the Author:

Neo and colleagues study the role of the Polycomb repressive complex 2 (PRC2) component EZH2 in yolk sac (YS) hematopoiesis. Through loss-of-functions studies the authors show that EZH2 function in YS endothelial cells essential for hematopoiesis. The authors show that in the absence of EZH2, purified erythro-myeloid progenitor cells (EMPs) fail to form any erythroid or myeloid colonies in vitro. The authors data also suggests that EZH2 ablation at later stages of hematopoiesis is dispensable for the generation of functional EMPs, although this part of analysis requires some additional work. Probing into molecular mechanisms the authors identified WNT signalling among the most highly enriched misregulated pathways. Finally, the authors show that inhibition of WNT signalling in EZH2-null EMPs can somewhat compensate for EZH2 loss in terms of EMPs functionality. It should be noted that somewhat similar gross observation for the requirement of PRC2 in early hematopoiesis was already made by Bing Zhang's group on 2017 (PMID: 28406475), although here authors pinpoint the role of PRC2/EZH2 to a much early developmental stage. The manuscript is well organized but currently lack several much-needed experiments to rigorously demonstrate these important findings, and to significantly advance our understandings of EZH2-mediated epigenetic regulation and the downstream mechanisms controlling YS hematopoiesis. Below are the specific comments.

1. Some clarification regarding the cre-deletion systems used in this study is required for the common reader who is not an expert in this system. Does both cre drivers get activated in YS EMPs, and more importantly at which embryonic day each of these drivers is being activated? Why E10.5 timepoint was specifically selected for analysis?

In line with these and the data shown in Fig.2a-b, how can the authors assure that at E10.5 Vav-Ezh2-KO EMPs are in fact EZH2 KO cells with no EZH2 protein (and downstream H3K27me3)? The tracing and RT-qPCR experiments in Fig1e-f does not address this concern adequately.

2. FACS profiles for Vav-Ezh2-KO and Tie2-Ezh2-KO EMPs isolation are missing. Also, data in Fig.1a-b shows absolute numbers – how changes in sample size were taken into an account? Authors should consider representing data as percentage of cells analysed.

3. Please define what are CFU- M, G, and E shown in CFU-C assays. There is no text description on how this assay was performed.

4. While authors nicely show that phenotypic EMPs are reduced in Tie2-Ezh2-KO mice at E10.5 and fail to form erythroid or myeloid colonies, the fate of these cells remains unclear. Does the in vivo reduction in Tie2-Ezh2-KO phenotypic EMPs increases over time? Does Tie2-Ezh2-KO EMPs fail to form colonies because they undergo apoptosis, senesce/arrest in differentiation program, or has skewed differentiation toward other cell types? Additional in vivo and in vitro experiments are warranted to address these open questions.

5. Which criteria were used for RNA-seq analysis? It seems that adjusted p-value was the only criteria used. If true, what is the biological meaning/significance for DE genes with expression levels lower than 1 FPKM in both WT and KO cells, or alternatively mild fold changes in expression when comparing to WT?

Authors should define a more rigorous criteria for DE genes and re-analyse data accordingly. Typically, genes with FPKM expression < 1, and genes with absolute fold change < 2 are not considered as DE genes.

6. How EZH2 target genes were determined in Fig.2c? There is no mentioning of EZH2 ChIP-seq data of EMPs in manuscript.

7. Perhaps authors can focus on direct EZH2 ChIP-seq targets instead of ATAC-seq data? What is the point that authors are trying to make in Fig.2d-f and how does it promote our understandings of the direct molecular mechanisms?

Since EZH2 is proposed to function as part of the Polycomb chromatin regulator, the focus should be on direct EZH2 target genes in EMPs. Also, if assuming EZH2 regulation is at the chromatin level, authors should also include H3K27me3 ChIP-seq data done in EMPs.

8. In lines 156-158 authors claim for a link between changes in gene expression and chromatin accessibility. However, authors did not provide any link between transcriptional changes and EZH2 function or EMPs functionality.

In other words, which genes are directly regulated by EZH2 in WT EMPs and from that set of genes which are upregulated in Tie2-Ezh2-KO EMPs?

9. In the experiments described by Fig.S3c outline, both D1 and D2 cells were analysed at D3, indicating that different time intervals were applied to D1 and D2 treated samples. Authors should therefore test the effect of EZH2 inhibition in D2 cultures at D4, giving similar treatment duration time to allow for downstream effects of inhibition to take place.

10. Is there a specific reason for different numbers of biological replicates within the various colony formation assays? Sample size varies from 3 to 13 in Fig.4e.

Authors should quantify additional biological replicates to reach at least somewhat similar sample sizes

analysed for each condition.

11. There are several additional (and apparently more prominent) pathways highlighted by GSEA analysis (RAS signalling, ER signalling). Can the authors explain why these pathways were not considered or any prior knowledge of these pathways which makes them less attractive direction? Especially since WNT inhibition seems to only partially revert the affect of EZH2 inhibition as shown in Fig.4c-d.

12. Following this line and related to point #4, apoptotic pathways are also highly enriched in GSEA analysis. How can authors be sure that inability to form colonies is not a consequence of cell death?

13. In line with previous comment, the link that authors made between EZH2 and WNT signalling is rather loose:

13A. Upregulated WNT genes mentioned by authors are in most cases poorly expressed in Tie2-Ezh2-KO EMPs with mean FPKM expression levels < 1 (Sulf2, Shisa3, etc...). Authors should repeat their analysis for enriched pathways after removing transcriptional noise and test whether WNT signalling remains a prominent pathway to follow.

13B. Regardless of changes in gene expression, authors should investigate if overall changes were in fact translated into aberrant activation of WNT signalling in Tie2-Ezh2-KO YS EMPs. The fact that WNT inhibition effects colony formation capacity, is not indicative that changes in WNT signalling took place in Tie2-Ezh2-KO EMPs.

13C. Since WNT inhibition also reduces EMPs colony formation potential, one would suspect that specific component(s) of this pathway are misregulated. Thus, in case the above suggested experiments demonstrate that WNT signalling is indeed hyper activated in Tie2-Ezh2-KO EMPs, authors should identify the direct EZH2 genes involved in its activation and perform functional assays to provide the detailed molecular mechanism.

14. Fig.4d-e data is missing analysis of CHIR99021 treatment with and without IWP2.

15. There are some inaccuracies in discussion lines 274-276. It is well known that H3K27me3 is mediated by either EZH1 or EZH2. While G9a could potentially provide another layer of gene repression by catalysing H3K9me3, it is not responsible for H3K27me3 histone modification per-se.

16. Previous findings of PRC2 roles in early haematopoiesis (PMID: 28406475) should be mentioned in introduction and discussion sections.

Reviewer #3:

Remarks to the Author:

In this manuscript, Neo et al study the role of Ezh2 during extra-embryonic yolk sac hematopoietic development. The authors claim that Ezh2 is an essential regulator of EMP hematopoiesis, and it accomplishes this by regulating Wnt signaling within nascent EMP. Overall, the studies performed by the authors are well-controlled, and provide new, intriguing insight into embryonic hematopoietic development. With a few additional experiments and clarifications, this study will provide a major impact.

Major concerns

1) the authors emphasize the role of Ezh2 on the EMP hematopoietic program. What about the program that precedes it, primitive hematopoiesis? Is there EryP-CFC potential within Tie2-Ezh2-KO E7.5 YS? Such studies could provide unique and complementary analyses into the nature of hemogenic endothelium and the primitive program, as well as strengthen the authors claim of an EMP-specific role for Ezh2.

- 2) The Wnt studies are performed exclusively in the background of chemical inhibition of Ezh2, with GSK126. The authors claims would be greatly strengthened if Wnt modulation were to be performed on Tie2-Ezh2-KO YS explants, to eliminate possible off-target effects from GSK126 treatment.
- 3) Can the authors comment on the effect of IWP2, a Porcn inhibitor? Does this suggest that Wnt is cell-autonomously downregulated by EMPs? Will a physiologically-relevant Wnt inhibition (Dkk, etc) similarly rescue EMP hematopoiesis?

Minor concerns

- 1) While the concentration of GSK126 was not indicated in Figure 4e, it appears that the same data from Figure 3a (10 uM GSK126) is being presented again.

Answers to reviewers

We thank both reviewers for their insightful and constructive comments. We have now addressed all their enquiries. In this document, we provide point-by-point responses. The corresponding changes in the revised manuscript have been highlighted in red. We hope that they will find our responses satisfying and will agree that the revised version of the manuscript has been substantially improved.

Reviewer #1 (Remarks to the Author):

This paper by Neo et al. is very interesting and relevant in that the authors clarified cell-intrinsic mechanisms that cause lethal anemia in Tie2-Ezh2-KO embryos, which has not been well understood despite that the severe anemia in Tie2-Ezh2-KO embryos was reported almost 10 years ago. The authors previously reported the cell-extrinsic role of Ezh2 in fetal liver hematopoiesis. In this paper, the authors examined the function of erythro-myeloid progenitors (EMP) in Tie2-Ezh2-KO YS. Surprisingly, they found complete diminishment of colony-forming ability in the YS, despite the presence of a normal number of phenotypic EMPs. This result suggested that EMPs were generated but not functional. Organ culture and using Ezh2 inhibitor, the authors demonstrated Ezh2 deletion did not alter endothelial-hematopoietic transition, but the hematopoietic colony-forming function. RNA- and ATAC-seq analysis identified the retention of EC features in Ezh2 deleted EMP and the upregulation of Wnt signaling. Finally, the authors tested the effect of upregulation of Wnt signaling in HECs using mouse ESC differentiation system and showed impairment of colony-forming ability when Wnt agonist was added to the HEC stage. Thus, this paper detailed the cause of impairment of EMPs in Tie2-Ezh2-KO embryos, which will be of great interest to the readers in the field of developmental hematopoiesis and epigenetics.

Authors Responses to Reviewer 1:

We greatly appreciate that the reviewer found merit in our manuscript, noting that i) it is "very interesting and relevant"; ii) We "detailed the cause of impairment of EMPs in Tie2-Ezh2-KO embryos" and iii) our paper "will be of great interest to the readers in the field of developmental hematopoiesis and epigenetics".

We also thank the reviewer for his/her constructive comments which have helped us to substantially improve our manuscript. Before addressing these comments in detail, we would first like to highlight a couple of the major revisions that we made in response to this reviewer's comments.

(1) We have characterized the primitive wave of hematopoiesis in Tie2-Ezh2-KO embryos and show that it is not affected by the loss of *Ezh2* via *Tie2-Cre*. This would suggest that the embryonic lethal anemic was likely to be the direct consequence of the non-functional Tie2-Ezh2-KO EMPs in combination with the cell-extrinsic hematopoietic impact of Ezh2 deletion in fetal liver endothelium.^{1,2}

(2) We have further characterized the Tie2-Ezh2-KO EMP and shown that they are proliferating and morphologically normal, not apoptotic or senescence. However, during the *in vitro* culture, Tie2-Ezh2-KO YS EMPs stop dividing after several rounds of divisions and became apoptotic.

Below we present the detailed answers to this reviewer's comments. Please note that we have highlighted in red in the revised manuscript the changes we made guided by all the reviewers' comments and suggestions.

There are a few questions.

1. The authors claimed that EHT is not altered, and phenotypical EMPs are generated, but they are not functional. How about the morphology of EMPs? I wonder if they are real blood cells or not. Do they look like EMPs when they are stained with Giemsa staining in cytopsin preparation?

We have performed the Wright-Giemsa staining on E10.5 YS EMP cytopsin and found no abnormality of the Tie2-Ezh2-KO EMPs in comparison to the WT or published images.³ The images are included in the Supplementary information, Fig. S3b.

2. Will EMPs from Tie2-Ezh2-KO embryos undergo apoptosis *in vitro* culture compared to WT?

We have performed Annexin V and 7AAD co-staining on E10.5 YS EMPs and found no significant difference between the percentage of apoptotic cells in WT and Tie2-Ezh2-KO YS EMPs (Supplementary information, Fig. S3c). However, during *in vitro* culture, Tie2-Ezh2-KO YS EMPs stopped dividing after several rounds of divisions and became apoptotic (Supplementary information, Fig. S3e; Video S1 and S2).

3. Are there any defects in primitive erythropoiesis at E7.5?

We thank the reviewer for raising this important point. We have now analyzed the primitive wave of hematopoiesis in the Tie2-Ezh2-KO embryos. We found similar levels of primitive erythroid (EryP, TER119⁺) in the Tie2-Ezh2-KO embryos than in their WT counterparts (Supplementary information, Fig. S1a, b). Tie2-Ezh2-KO embryos also exhibited similar clonogenic potential than WT (Supplementary information, Fig. S1c) and in particular generated EryP colonies (Supplementary information, Fig. S1d). Altogether, these results indicate that primitive hematopoiesis is not affected in Tie2-Ezh2-KO.

4. In Fig. S3d FACS plots, the gating of HE1 looks CD41 dim, not negative. Please show FMO to show HE1 is CD41 negative.

We have added the FMO control in the Supplementary information, Fig. S7d and redefined the CD41⁻ as CD41^{lo}.

--

Reviewer #2 (Remarks to the Author):

Neo and colleagues study the role of the Polycomb repressive complex 2 (PRC2) component EZH2 in yolk sac (YS) hematopoiesis. Through loss-of-functions studies the authors show that EZH2 function in YS endothelial cells essential for hematopoiesis. The authors show that in the absence of EZH2, purified erythro-myeloid progenitor cells (EMPs) fail to form any erythroid or myeloid colonies *in vitro*. The authors data also suggests that EZH2 ablation at later stages of hematopoiesis is dispensable for the generation of functional EMPs, although this part of analysis requires some additional work. Probing into molecular mechanisms the authors identified WNT signalling among the most highly enriched misregulated pathways. Finally, the authors show that inhibition of WNT signalling in EZH2-null EMPs can somewhat compensate for EZH2 loss in terms of EMPs functionality. It should be noted that somewhat similar gross observation for the requirement of PRC2 in early hematopoiesis was already made by Bing Zhang's group on 2017 (PMID: 28406475), although here authors pinpoint the role of PRC2/EZH2 to a much early developmental stage. The manuscript is well organized but currently lack several much-needed experiments to rigorously demonstrate these important findings, and to significantly advance our understandings of EZH2-mediated epigenetic regulation and the downstream mechanisms controlling YS hematopoiesis. Below are the specific comments.

Authors Responses to Reviewer 2:

We are pleased to read that the reviewer found merit in our manuscript, noting that i) We "pinpoint the role of PRC2/EZH2 to a much early developmental stage" and ii) our paper "is well organized". We also thank the reviewer for his/her constructive comments which have helped us to substantially improve our manuscript. Before addressing these comments in detail, we would first like to highlight a couple of the major revisions that we made in response to this reviewer's comments.

(1) We performed H3K27me3 CUT&RUN analysis on E10.5 YS EMP to identify the target genes of EZH2. EZH2 CUT&RUN was extremely technically challenging for a rare cell population such as EMPs. We have identified that a large proportion of the EC, HE, and Wnt signaling genes were directly targeted by EZH2 based on the presence of H3K27me3 marks.

(2) We have further characterized the Tie2-Ezh2-KO EMP and shown that they are proliferating and morphologically normal, are not apoptotic or senescent. However, upon *in vitro* culture, Tie2-Ezh2-KO YS EMPs stop dividing after several rounds of divisions and become apoptotic.

Below we present the detailed answers to this reviewer's comments. Please note that we have highlighted in red in the revised manuscript the changes we made guided by all the reviewers' comments and suggestions.

1. Some clarification regarding the cre-deletion systems used in this study is required for the common reader who is not an expert in this system. Does both cre drivers get activated in YS EMPs, and more importantly at which embryonic day each of these drivers is being activated? Why E10.5 timepoint was specifically selected for analysis?

We apologize for not explaining the Cre-deletion systems in the manuscript clearly. Tie2 (tunica intima endothelial kinase 2/Tek) is a receptor tyrosine kinase that binds angiopoietin-1 and angiopoietin-2. Tie2 is expressed in all endothelial cells,⁴ alongside hematopoietic cells in the AGM region, fetal liver and adult bone marrow.^{5,6} Previous studies with this mouse strain revealed the presence of Tie2 positive mesodermal precursors at E7.5 and endothelial and hematopoietic cells at

E8.5 in yolk sacs and the dorsal aorta.⁴ Based on our Cre-loxP (Rosa26-LSL-tdTomato) reporter system to trace the progeny of Tie2⁺ cells in distinct lineages of the hematopoietic system, we found that nearly all EMPs are the progeny of Tie2⁺ cells (Fig. 1e). Together, these data suggest that Tie2-Cre is activated in the E8.5 endothelium where the generation of EMPs is initiated. In contrast, Vav1, a guanine nucleotide exchange factors (GEFs) for Rho family GTPases, is expressed virtually in all hematopoietic cells and this is closely reflected by its iCre activity.⁷ Vav1 expression only becomes prominent in the EMP stage (see our single-cell RNA-seq data below). This is in line with our Cre-LoxP reporter analysis which indicating a complete activation of Cre in EMPs (Fig. 1e). In addition, the RT-qPCR (Fig. 1f) and ATAC-seq (Supplementary information, Fig. S6b) also strongly suggest a progressive loss of EZH2 activity in Vav-Ezh2-KO EMPs compared to the Tie2-Ezh2-KO. Overall, Vav1 expression is subsequent to Tie2 during EHT, and therefore both represent valuable tools to dissect the requirement of Ezh2 for EMP generation at different stages of EHT.

We decided to analyze the EMP mainly at E10.5 based on several reasons. First, even if EMPs emerge in the YS from E8.5 until as late as E11.5,^{8,9} they are more homogenous and abundant at E10.5 compared to E9.5. Secondly, E10.5 is the latest time point we can examine these EMPs before the majority of them migrate from the YS to embryo proper via blood circulation. Finally, as the epigenetic factor Ezh2 requires time to establish repressive histone marks on the chromatin, potential effects arising from Ezh2 inactivation are more likely to be prominent at a later rather than an earlier timepoint.

In line with these and the data shown in Fig.2a-b, how can the authors assure that at E10.5 Vav-Ezh2-KO EMPs are in fact EZH2 KO cells with no EZH2 protein (and downstream H3K27me3)? The tracing and RT-qPCR experiments in Fig1e-f does not address this concern adequately.

We thank the reviewer for raising this important point. Our Ezh2 floxed model deletes part of the SET domain (Ex16-Ex19 of Ex14-Ex19) of Ezh2.¹⁰ It has been shown that this EZH2 Δ SET mutant has significantly reduced methyltransferase activity.¹⁰ It is technically challenging to measure EZH2 protein level within YS EMPs due to their scarcity (1000-2000 cells/YS). Lack of reliable antibody that can distinguish the EZH2 Δ SET from WT EZH2 also prohibits the analysis of their protein level via immunofluorescence staining. However, we agree that the complete loss of the WT EZH2 protein is unlikely at an early stage of EMPs when the deletion of the SET domain is only occurring at the genomic level. The replacement of WT EZH2 with EZH2 Δ SET protein should probably be progressive and take time to complete. Nevertheless, comparing the Tie2-Ezh2-KO and Vav-Ezh2-KO phenotype provide valuable insight toward the requirement of EZH2 at different stages of EHT and EMP generation. Our RT-qPCR (Fig. 1f) and ATAC-seq confirm a delayed deletion of Ezh2 in Vav-Ezh2-KO EMPs compared to the Tie2-Ezh2-KO EMPs (Supplementary information, Fig. S6b). Furthermore, the different consequences of EZH2 inhibitor treatments of mESC derived HE1, HE2 and HP populations also indicate different requirements for EZH2 at different stages of EHT (Fig. 3f).

2. FACS profiles for Vav-Ezh2-KO and Tie2-Ezh2-KO EMPs isolation are missing. Also, data in Fig.1a-b shows absolute numbers – how changes in sample size were taken into an account? Authors should consider representing data as percentage of cells analysed.

We have now included the FACS gating strategy we used to isolate the YS EMPs (Supplementary information, Fig. S2). We also modified our data in Fig. 1a, b to show the percentages of EMPs.

3. Please define what are CFU- M, G, and E shown in CFU-C assays. There is no text description on how this assay was performed.

We apologize for not explaining the CFU assay in detail in our manuscript. We have added additional descriptions in the method section to provide further explanation on the CFU assay.

CFU assay was performed by culturing cells in a semi-solid methylcellulose matrix with appropriate cytokines and supplements (M3434, Stem Cell Technologies). Discrete cell clusters or colonies were counted after 7-10 days of culture. Types of colony were defined according to their distinct morphology. CFU-G/M, Colony-forming unit-granulocyte/macrophage; CFU-G/M/E, Colony-forming unit-granulocyte/macrophage/erythroid; CFU-E, Colony-forming unit-erythroid.

4. While authors nicely show that phenotypic EMPs are reduced in Tie2-Ezh2-KO mice at E10.5 and fail to form erythroid or myeloid colonies, the fate of these cells remains unclear. Does the in vivo reduction in Tie2-Ezh2-KO phenotypic EMPs increases over time? Does Tie2-Ezh2-KO EMPs fail to form colonies because they undergo apoptosis, senesce/arrest in differentiation program, or has skewed differentiation toward other cell types? Additional in vivo and in vitro experiments are warranted to address these open questions.

The quantification of total EMPs number beyond E10.5 would be technically difficult as the EMPs start to colonize different tissues. However, measuring the numbers of EMPs in the fetal liver, the major organ colonized by the EMPs, at E11.5 and E12.5, suggested that the proliferation of Tie2-Ezh2-KO EMPs is comparable to the WT despite a clear reduction in initial number of cells (Supplementary information, Fig. S3a). We have performed Annexin V / 7AAD staining on E10.5 YS EMPs and found no significant difference in the percentage of apoptotic cells between WT and Tie2-Ezh2-KO YS EMPs (Supplementary information, Fig. S3c). However, upon *in vitro* culture, Tie2-Ezh2-KO YS EMPs stopped dividing after several rounds of division and became apoptotic (Supplementary information, Fig. S3e; Video S1 and S2). GSEA and the absence of change in *Il6* expression suggest that the Tie2-Ezh2-KO YS EMPs do not senesce (Supplementary information, Fig. S3d). To address the potential changes in lineage bias of Tie2-Ezh2-KO EMPs, we analyzed their gene expression of myeloid, lymphoid, erythroid and platelet lineage-specific genes and found no pronounced lineage skewing compared to the WT EMPs (Supplementary information, Fig. S4b).

5. Which criteria were used for RNA-seq analysis? It seems that adjusted p-value was the only criteria used. If true, what is the biological meaning/significance for DE genes with expression levels lower than 1 FPKM in both WT and KO cells, or alternatively mild fold changes in expression when comparing to WT? Authors should define a more rigorous criteria for DE genes and re-analyse data accordingly. Typically, genes with FPKM expression < 1, and genes with absolute fold change < 2 are not considered as DE genes.

We apologize for not stating clearly the filtering criteria we applied in our studies. We only applied adjusted p-value ($p > 0.05$) for our initial differential expressed gene (DEG) analysis. We agree with the reviewer that removing the lowly expressed genes would help to reduce the background noise of the analysis. Lowly expressed genes were determined based on the RPKM distribution. The dynamic range of gene expression across all samples as measured in RPKM is shown below. The mean expression of individual genes ($n=16,876$) is shown as grey dots. The shaded area of the box plot indicates the upper- (75%) and lower- (25%) percentile of the RPKM value. The density plot above the box plot shows the distribution of expression value. Based on the RPKM distribution, the typical RPKM filtering criteria is not applicable in this cell type where it would remove 44% ($n=7425$) of the total genes and make the subsequent analysis not statistically reliable. A cut-off of mean RPKM 0.2 was deemed more suitable. The cut-off of 0.2 was one-fold higher than the 25% percentile of the data (0.11 RPKM). Genes with a mean count of < 0.2 across all cells were removed. A total of 14,393 genes (85%) had mean counts of more than 0.2 and were retained. However, despite the low mRNA content in our samples, having 5 biological replicates for each population undoubtedly increased the confidence of our dataset on detecting statistically significant changes in gene expression. Indeed, our DEG lists are in line with our ATAC-seq and H3K27me3 CUT&RUN data, further strengthen the reliability of our dataset. We have reanalysed all our data with the filtered RPKM gene list.

6. How EZH2 target genes were determined in Fig.2c? There is no mentioning of EZH2 ChIP-seq data of EMPs in manuscript.

We apologize for not stating the dataset (GSEA Gene Set: PRC2_EZH2_UP.V1_UP) that we used to define the Ezh2 target genes in our original manuscript. We agree that an EZH2 ChIP-seq would be ideal to define the EZH2 target genes in YS EMPs. However, the requirement of millions of cells for a ChIP-seq is technically impossible with the rare EMP population (1,000-2,000/YS). Therefore, we have employed the new CUT&RUN (Cleavage Under Targets & Release Using Nuclease) method which only requires hundreds to thousands of cells to identify the EZH2 target genes in YS EMPs. Despite multiple attempts trying to optimize an EZH2 CUT&RUN, we failed to obtain a reliable result with over 10,000 EMPs. Although it might be possible to improve results by increasing the cell numbers, this is unfortunately not feasible with the limited EMPs. Therefore, we decided to infer the EZH2 target genes in EMPs by performing H3K27me3 CUT&RUN. We successfully obtained reliable results with 500 and 15,000 E10.5 YS EMPs (Supplementary information, Table S2). We inferred the EZH2 target genes base on the presence of H3K27me3. Overall, we detected 5,352 peaks which corresponded to 2,420 potential EZH2 target genes (500 EMPs; ± 3 kb from TSS; $p < 0.01$) 7,093 peaks which corresponded to 4,129 potential EZH2 target genes (15,000 EMPs; ± 3 kb from TSS; $p < 0.05$). We increased the p-value stringency for the 500 EMPs dataset due to the inherent high background noise with a low cell number sample. Nearly 99% of the potential EZH2 targets identified from the 500 EMPs were present in the 15,000 EMPs H3K27me3 CUT&RUN (Supplementary information, Fig S5a). This result suggest that these data are highly reliable.

7. Perhaps authors can focus on direct EZH2 ChIP-seq targets instead of ATAC-seq data? What is the point that authors are trying to make in Fig.2d-f and how does it promote our understandings of the direct molecular mechanisms? Since EZH2 is proposed to function as part of the Polycomb chromatin regulator, the focus should be on direct EZH2 target genes in EMPs. Also, if assuming EZH2

regulation is at the chromatin level, authors should also include H3K27me3 ChIP-seq data done in EMPs.

We completely agree with the reviewer that EZH2 ChIP-seq would be an obvious direct approach to study the molecular regulatory mechanism of Ezh2 in EMPs. However, due to the limitation in EMP cell number, the conventional ChIP-seq approach that requires millions of cells is not technically feasible. Therefore, we have decided to perform low cell number ATAC-seq with 500 EMPs as an indirect way to infer the potential Ezh2 regulated genes. Indeed, our ATAC-seq data correlate nicely with the RNA-seq data (Fig. 2f, g; Supplementary information, Fig. S6). Since we submitted our manuscript, a new chromatin profiling technique (CUT&RUN) has become more popular. CUT&RUN uses a target-specific primary antibody and a Protein A-Protein G-Micrococcal Nuclease (pAG-MNase) to isolate specific protein-DNA complexes.¹¹⁻¹³ With CUT&RUN, one can profiling the chromatin with less than 100 cells for a histone modification and 1,000 cells for a transcription factor (TF).¹² However, the requirement of cell number varies according to the expression/abundance of TFs or histone marks. As such, we tried to optimize the protocol for EZH2 with over 10,000 EMPs but failed to generate reliable results. Therefore, we focused on H3K27me3 CUT&RUN and successfully acquired high-quality data with just 500 and 15,000 EMPs (Supplementary information, Fig. S5a). We inferred the EZH2 target genes base on the presence of H3K27me3. Overall, we detected 5,352 peaks which corresponded to 2,420 potential EZH2 target genes (500 EMPs; ± 3 kb from TSS; $p < 0.01$) 7,093 peaks which corresponded to 4,129 potential EZH2 target genes (15,000 EMPs; ± 3 kb from TSS; $p < 0.05$). In line with our RNA-seq data, a large proportion of EC (33.3%), HE (34%) and Wnt (45.2%) signaling genes were EZH2 targets but not with the hematopoietic specific genes (5.8%) (Fig. 2d, 4b; Supplementary information, Table S4). These data strongly suggest that Ezh2 is directly regulating the expression of a large subset of EC, HE and Wnt signaling genes in YS EMPs which need to be downregulated at the end of the EHT.

8. In lines 156-158 authors claim for a link between changes in gene expression and chromatin accessibility. However, authors did not provide any link between transcriptional changes and EZH2 function or EMPs functionality. In other words, which genes are directly regulated by EZH2 in WT EMPs and from that set of genes which are upregulated in Tie2-Ezh2-KO EMPs?

We have performed H3K27me3 CUT&RUN in E10.5 YS EMPs to infer the EZH2 target genes. Overall, we detected 5,352 peaks which corresponded to 2,420 potential EZH2 target genes (500 EMPs; ± 3 kb from TSS; $p < 0.01$) or 7,093 peaks which corresponded to 4,129 potential EZH2 target genes (15,000 EMPs; ± 3 kb from TSS; $p < 0.05$). We increased the p-value stringency for the 500 EMPs dataset due to the inherent high background noise with a low cell number sample. Nearly 99% of the potential EZH2 targets identified from the 500 EMPs were present in the 15,000 EMPs H3K27me3 CUT&RUN (Supplementary information, Fig S5a). As the lower input number (500 EMPs) probably limits the detection of EZH2 targets, we considered all the targets identified with 15,000 EMPs as potential EZH2 target genes (Supplementary information, Table S3, S4). The majority of the EZH2 target genes were indeed upregulated and enriched in the Tie2-Ezh2-KO EMPs (55%; Fig. 2c). A large proportion of EC and HE enriched genes were targets of EZH2, unlike the hematopoietic genes, a finding in line with their upregulation Tie2-Ezh2-KO EMPs detected by GSEA (Fig. 2d; Supplementary information, Table S3). This is further supported by the gene ontology (GO) analysis which revealed a significant

enrichment of EC-related genes (Cadherin signaling pathway and Angiogenesis) in the EZH2 target genes (Supplementary information, Fig. S5b).

9. In the experiments described by Fig.S3c outline, both D1 and D2 cells were analysed at D3, indicating that different time intervals were applied to D1 and D2 treated samples. Authors should therefore test the effect of EZH2 inhibition in D2 cultures at D4, giving similar treatment duration time to allow for downstream effects of inhibition to take place.

We apologize for not having described better the rationale of our experimental design. The key question we are asking is at which developmental stages of EHT is EZH2 enzymatic activity required. During the course of EHT culture, the cell populations are relatively synchronous. Based on our previous studies, the cultures are enriched for hemangioblast/HE1 on day 1, HE1/HE2 on day 2 and hematopoietic progenitors on day 3. Therefore, by inhibiting the EZH2 activity on day 1 or day 2, we are examining the effects of blocking EZH2 activity on EHT at early or late stage of EHT respectively. We are replating for colony assays (CFU) in methylcellulose at day 3 as our previous studies have indicated that it is the optimal time, as these culture conditions do not support hematopoietic progenitors for an additional day. However, we are adding the EZH2 inhibitor in methylcellulose during the CFU-C assay. The total duration of the EZH2 inhibition would be therefore relatively similar. In addition, our study suggests that the key role of Ezh2 during EHT is to establish repressive histone marks at the late phase of EHT. Therefore, the duration of EZH2 inhibition should matter less as we are looking at the impact of blocking the establishment rather than depletion of the H3K27me3 mark.

10. Is there a specific reason for different numbers of biological replicates within the various colony formation assays? Sample size varies from 3 to 13 in Fig.4e. Authors should quantify additional biological replicates to reach at least somewhat similar sample sizes analysed for each condition.

We have modified the Fig. 4e to present similar biological replicates. We have also included new data acquired with the same drug treatments of E10.5 YS EMP to eliminate potential off-target effects of the EZH2 inhibitor, as suggested by the other reviewer. These new data are now replacing the previous figure and presented as Fig. 4g in the revised manuscript.

11. There are several additional (and apparently more prominent) pathways highlighted by GSEA analysis (RAS signalling, ER signalling). Can the authors explain why these pathways were not considered or any prior knowledge of these pathways which makes them less attractive direction? Especially since WNT inhibition seems to only partially revert the affect of EZH2 inhibition as shown

We agree with the reviewer that analyzing other pathways (RAS and ER signaling) would be indeed very informative. However, it's beyond the scope of the current paper to study multiple pathways in parallel. The reason we decided to focus on Wnt signaling was because it was already previously associated with EHT. Indeed, a previous study has indicated the essential role of Wnt signaling in the early stage of EHT for YS EMP generation.⁹

12. Following this line and related to point #4, apoptotic pathways are also highly enriched in GSEA analysis. How can authors be sure that inability to form colonies is not a consequence of cell death?

We have performed the Annexin V and 7AAD staining on the E10.5 YS EMPs and found no significant difference between the percentage of apoptotic cells between the WT and Tie2-Ezh2-KO YS EMPs (Supplementary information, Fig. S3c). However, during *in vitro* culture, Tie2-Ezh2-KO YS EMPs stopped dividing after several rounds of divisions and became apoptotic (Supplementary information, Fig. S3e; Video S1 and S2).

13. In line with previous comment, the link that authors made between EZH2 and WNT signalling is rather loose:

13A. Upregulated WNT genes mentioned by authors are in most cases poorly expressed in Tie2-Ezh2-KO EMPs with mean FPKM expression levels < 1 (Sulf2, Shisa3, etc...). Authors should repeat their analysis for enriched pathways after removing transcriptional noise and test whether WNT signalling remains a prominent pathway to follow.

We appreciate the concern raised by the reviewer. We have replaced Fig. 4a with a new GSEA performed with the filtered RPKM data ($P < 0.05$, $RPKM > 0.2$) and the enrichment remains significant ($P = 0.012$). As we have five biological replicates for each group and found a significant difference in the expression of these genes, we believe that they represent real biological changes in Wnt signaling. This is now further supported by the increased nuclear β -CATENIN in the Tie2-Ezh2-KO EMPs (Fig. 4c). A previous study has already suggested changes in Wnt signaling over the EHT process.⁹ Accordingly, our results with Wnt activator/inhibitor treatment of either *ex vivo* EMP culture or mESC EHT culture (Fig. 4d-g) demonstrated their sensitivity on Wnt signaling dosage and confirm the functional link between Wnt signaling and Ezh2.

13B. Regardless of changes in gene expression, authors should investigate if overall changes were in fact translated into aberrant activation of WNT signalling in Tie2-Ezh2-KO YS EMPs. The fact that WNT inhibition effects colony formation capacity, is not indicative that changes in WNT signalling took place in Tie2-Ezh2-KO EMPs.

We have performed immunostaining on the sorted E10.5 YS EMPs and observed increased nuclear β -CATENIN in the Tie2-Ezh2-KO EMPs (Fig. 4c). These data are consistent with activated Wnt signaling in the Tie2-Ezh2-KO EMPs.

13C. Since WNT inhibition also reduces EMPs colony formation potential, one would suspect that specific component(s) of this pathway are misregulated. Thus, in case the above suggested experiments demonstrate that WNT signalling is indeed hyper activated in Tie2-Ezh2-KO EMPs, authors should identify the direct EZH2 genes involved in its activation and perform functional assays to provide the detailed molecular mechanism.

We agree with the reviewer that identifying the specific Wnt signaling genes that are responsible for the phenotype would be interesting. Therefore, we performed H3K27me3 CUT&RUN to identify

EZH2 target genes. We detected 7093 peaks which corresponded to 4129 potential EZH2 target genes (\pm 3kb from TSS; $P < 0.05$; Supplementary information, Table. S2). Furthermore, we found that 19 out of 42 Wnt signaling pathway genes (Hallmark gene set) are potential EZH2 targets (Supplementary information, Table S4). Among these 19 genes, 7 (*Tcf7*, *Jag1*, *Jag2*, *Gnai1*, *Hdac11*, *Ccnd2*, *Nkd1*) were significantly upregulated in Tie2-Ezh2-KO EMPs (Supplementary information, Table S1). To further dissect the precise role of these individual potential EZH2's Wnt target genes, extensive genetic modifications would have to be made in the EMP population. This would be labor-intensive and technically challenging due to the scarcity of EMP. Interestingly, when we expanded the list of Wnt signaling pathways gene based on literature, we also found the expression of *Wnt4* and *Wnt5b* ligands alongside increased *Porcn* expression in Tie2-Ezh2-KO EMPs (Supplementary information, Fig. S9). Importantly, all these three genes are potentially EZH2 targets (Supplementary information, Fig. S9). PORCN plays an essential role in processing WNT ligands via palmitoylation to modulate their secretion.¹⁴ IWP2, a PORCN inhibitor, selectively blocks the palmitoylation of WNT ligands by PORCN.¹⁵ We observed that IWP2 treatment on mESC EHT cultures partially rescues the phenotype caused by the EZH2 inhibition (Fig. 4d-f). These results suggest that WNT ligands secretion from either the niche populations or EMP itself are playing a role in regulating the maturation of EMPs. The fact that similar rescue was also observed with the treatment on Tie2-Ezh2-KO EMPs would indicate that EMPs do express and secreting the WNT ligands (Fig. 4g). Together, this would suggest that Ezh2 is regulating Wnt activity by repressing the expression of PORCN which is required for the processing and secretion of WNT4 and WNT5b ligands.

In our manuscript, we have shown that the absence of Ezh2 induced aberrant Wnt signaling resulting in non-functional EMP. To further strengthen the link between the Wnt signaling and EMP functionality, we have also now treated the Tie2-Ezh2-KO E10.5 YS EMPs with another type of Wnt inhibitor (DKK1) that sequestered the LRP6 co-receptor that prevents activation of the Wnt signaling pathway upon ligand binding.^{16,17} This is different from IWP2 which blocks the Wnt signaling by preventing the secretion of WNT ligands. The fact that we observed a similar rescue effect with DKK1 further support the importance of regulated Wnt activity in EMP functionality and rule out the potential off-target effects of IWP2.

14. Fig.4d-e data is missing analysis of CHIR99021 treatment with and without IWP2.

We thank the reviewer for pointing this out. However, it would be hard to interpret the outcome of the treatment with both Wnt activator (CHIR99021) and inhibitor (IWP2) together as it will be mainly driven by their respective dosage effect. Instead, we have performed the CHIR99021 treatment together with the EZH2 inhibitor (GSK126) and included the new data in Fig. 4d-f. Overall, additional Wnt inhibition does not have any synergistic or additive effect alongside EZH2 inhibition. This is further supported by the CFU-C assay on CHIR99021 treated Tie2-Ezh2-KO EMPs, where no further reduction of hematopoietic colony output was observed (Fig. 4g).

15. There are some inaccuracies in discussion lines 274-276. It is well known that H3K27me3 is mediated by either EZH1 or EZH2. While G9a could potentially provide another layer of gene repression by catalysing H3K9me3, it is not responsible for H3K27me3 histone modification per-se. We agree that G9a has only been shown to contribute *in vitro* and *in vivo* towards H3K27me1/2

methylation.¹⁸⁻²⁰ There are also data suggesting G9a might facilitate the trimethylation of H3K27 by interacting with Ezh2.²¹ However, we agree with the reviewer that the link between G9a and H3K27me3 is not well established. Therefore, we decided to remove the section from our discussion.

16. Previous findings of PRC2 roles in early haematopoiesis (PMID: 28406475) should be mentioned in introduction and discussion sections.

The paper mentioned by the reviewer has now been added in the introduction of our manuscript. This paper revealed the reduction of phenotypic FL HSC upon the depletion of EED with Tie2-Cre. Mochizuki-Kashio et al. observed a similar decrease with a Tie2-Ezh2-KO model.²² Although the reduction of phenotypic HSC does translate into lower engraftment of these Tie2-Ezh2-KO HSC, the existence of engraftable Tie2-Ezh2-KO HSC would argue against an intrinsic functional defect. Indeed, we previously demonstrated the presence in Tie2-Ezh2-KO embryos of functional FL HSC by FACS and transplantation assays alongside the expected emergence of HSPC in the AGM.¹ We proposed that the reduction of functional HSC is mainly due to the lack of membrane-bound KitL in the FL niche, critical for HSC expansion, rather than an intrinsic defect of Tie2-Ezh2-KO HSC. The dispensable role of Ezh2 in FL HSC is further supported by Xie et al.²³ and our observation¹ that Vav-Ezh2-KO, specifically targeting hematopoietic cells, has normal levels of FL HSC by FACS and transplantation assay. Together, these data would suggest a dispensable intrinsic role of Ezh2 for functional fetal HSC.

--

Reviewer #3 (Remarks to the Author):

In this manuscript, Neo et al study the role of Ezh2 during extra-embryonic yolk sac hematopoietic development. The authors claim that Ezh2 is an essential regulator of EMP hematopoiesis, and it accomplishes this by regulating Wnt signaling within nascent EMP. Overall, the studies performed by the authors are well-controlled, and provide new, intriguing insight into embryonic hematopoietic development. With a few additional experiments and clarifications, this study will provide a major impact.

Authors Responses to Reviewer 3:

We greatly appreciate that the reviewer found merit in our manuscript, noting that our study i) We "are well-controlled, and provide new, intriguing insight into embryonic hematopoietic development" and ii) our paper has the potential to "provide a major impact". We also thank the reviewer for his/her constructive comments which have helped us to substantially improve our manuscript.

We also thank the reviewer for his/her constructive comments which have helped us to substantially improve our manuscript. Before addressing these comments in detail, we would first like to highlight a couple of the major revisions that we made in response to this reviewer's comments.

(1) We have characterized the primitive wave of hematopoiesis in Tie2-Ezh2-KO embryos and show that it is not affected by the loss of *Ezh2* via *Tie2-Cre*. This would suggest that the embryonic lethal anemic was likely to be the direct consequence of the non-functional Tie2-Ezh2-KO EMPs in combination with the cell-extrinsic hematopoietic impact of Ezh2 deletion in fetal liver endothelium.^{1,2}

(2) We have further expanded our study on Ezh2 regulated Wnt signaling in EMP generation by including another type of Wnt inhibitor (DKK1) and performed the experiment with purified EMPs to avoid potential off-target effects from the EZH2 inhibitor.

We present below detailed answers to this reviewer's comments. Please note that we have highlighted in red in the revised manuscript the changes we made guided by all the reviewers' comments and suggestions.

Major concerns

1) The authors emphasize the role of Ezh2 on the EMP hematopoietic program. What about the program that precedes it, primitive hematopoiesis? Is there EryP-CFC potential within Tie2-Ezh2-KO E7.5 YS? Such studies could provide unique and complementary analyses into the nature of hemogenic endothelium and the primitive program, as well as strengthen the authors claim of an EMP-specific role for Ezh2.

We thank the reviewer for raising this important point. We have now analyzed the primitive wave of hematopoiesis in the Tie2-Ezh2-KO embryos. We found similar levels of primitive erythroid (EryP, TER119⁺) cells in the Tie2-Ezh2-KO embryos than in their WT counterparts (Supplementary information, Fig. S1a, b). Tie2-Ezh2-KO embryos also exhibited normal clonogenic potential (Supplementary information, Fig. S1c) and generated EryP colonies (Supplementary information, Fig. S1d). Altogether, these results indicate that primitive hematopoiesis is not affected in Tie2-Ezh2-KO.

2) The Wnt studies are performed exclusively in the background of chemical inhibition of Ezh2, with GSK126. The authors claims would be greatly strengthened if Wnt modulation were to be performed on Tie2-Ezh2-KO YS explants, to eliminate possible off-target effects from GSK126 treatment.

We thank the reviewer for this suggestion. We have now replaced Fig. 4g (Fig. 4e in initial manuscript) with the data acquired from the treatment of Wnt activator (CHIR99021) or inhibitor (IWP2 or DKK1) on FACS sorted E10.5 YS EMP. With these new data, we not only rule out the potential off-target effect of GSK126 treatment as the reviewer suggested, but also potential effect from the niche. Indeed, we reproduce our previous results that indicated that EMP functionality is sensitive to the level of Wnt signaling and that Wnt inhibition can partially rescue the defect of Tie2-Ezh2-KO EMPs.

3) Can the authors comment on the effect of IWP2, a Porcn inhibitor? Does this suggest that Wnt is cell-autonomously downregulated by EMPs? Will a physiologically-relevant Wnt inhibition (Dkk, etc) similarly rescue EMP hematopoiesis?

IWP2, a PORCN inhibitor, selectively blocks the palmitoylation of WNT ligands by PORCN which is required for their secretion.¹⁵ We observed that IWP2 treatment on the mESC EHT culture partially rescue the phenotype caused by the EZH2 inhibition (Fig. 4d-f). These findings suggest that WNT ligands secretion from either the niche populations or EMP itself are playing a role in regulating the maturation of EMPs. The fact that similar rescue was also observed with the treatment on purified Tie2-Ezh2-KO EMPs suggest that EMPs do express and secrete WNT ligands (Fig. 4g). This is supported by our RNA-seq data which show expression of *Wnt4* and *Wnt5b* ligands alongside increased *Porcn* expression in Tie2-Ezh2-KO EMPs (Supplementary information, Fig. S9). Importantly, all these three genes are potentially EZH2 targets (Supplementary information, Fig. S9). Altogether, this would suggest that Ezh2 is regulating the Wnt activity by repressing the expression of PORCN which is required for the processing and secretion of WNT4 and WNT5b ligands.

As suggested by the reviewer, we have tested another Wnt inhibitor (DKK1) which sequestered the LRP6 co-receptor, preventing activation of the Wnt signaling pathway upon ligand binding.^{16,17} This is different from IWP2 which blocks the Wnt signaling by preventing the secretion of WNT ligand. We observed a similar rescue with DKK1, further supporting the importance of regulated Wnt activity in EMP functionality and ruling out potential off-target effects of IWP2.

Minor concerns

1) While the concentration of GSK126 was not indicated in Figure 4e, it appears that the same data from Figure 3a (10 uM GSK126) is being presented again.

We apologize for this mistake. We have now replaced the Fig. 4e (Fig. 4g in the revised manuscript) with new data and indicated the drug concentration for each compound in the figure legend.

References

- 1 Neo, W. H. *et al.* Cell-extrinsic hematopoietic impact of Ezh2 inactivation in fetal liver endothelial cells. *Blood* **131**, 2223-2234, doi:10.1182/blood-2017-10-811455 (2018).
- 2 Azzoni, E. *et al.* Kit ligand has a critical role in mouse yolk sac and aorta-gonad-mesonephros hematopoiesis. *EMBO Rep* **19**, doi:10.15252/embr.201745477 (2018).
- 3 Mass, E. *et al.* Specification of tissue-resident macrophages during organogenesis. *Science* **353**, doi:10.1126/science.aaf4238 (2016).
- 4 Kisanuki, Y. Y. *et al.* Tie2-Cre transgenic mice: a new model for endothelial cell-lineage analysis in vivo. *Dev Biol* **230**, 230-242, doi:10.1006/dbio.2000.0106 (2001).
- 5 Puri, M. C. & Bernstein, A. Requirement for the TIE family of receptor tyrosine kinases in adult but not fetal hematopoiesis. *Proc Natl Acad Sci U S A* **100**, 12753-12758, doi:10.1073/pnas.2133552100 (2003).
- 6 Takakura, N. *et al.* Critical role of the TIE2 endothelial cell receptor in the development of definitive hematopoiesis. *Immunity* **9**, 677-686 (1998).
- 7 de Boer, J. *et al.* Transgenic mice with hematopoietic and lymphoid specific expression of Cre. *Eur J Immunol* **33**, 314-325, doi:10.1002/immu.200310005 (2003).
- 8 McGrath, K. E. *et al.* Distinct Sources of Hematopoietic Progenitors Emerge before HSCs and Provide Functional Blood Cells in the Mammalian Embryo. *Cell Rep* **11**, 1892-1904, doi:10.1016/j.celrep.2015.05.036 (2015).
- 9 Frame, J. M., Fegan, K. H., Conway, S. J., McGrath, K. E. & Palis, J. Definitive Hematopoiesis in the Yolk Sac Emerges from Wnt-Responsive Hemogenic Endothelium Independently of Circulation and Arterial Identity. *Stem Cells* **34**, 431-444, doi:10.1002/stem.2213 (2016).
- 10 Su, I. H. *et al.* Ezh2 controls B cell development through histone H3 methylation and Igh rearrangement. *Nat Immunol* **4**, 124-131, doi:10.1038/ni876 (2003).
- 11 Skene, P. J. & Henikoff, S. An efficient targeted nuclease strategy for high-resolution mapping of DNA binding sites. *Elife* **6**, doi:10.7554/eLife.21856 (2017).
- 12 Skene, P. J., Henikoff, J. G. & Henikoff, S. Targeted in situ genome-wide profiling with high efficiency for low cell numbers. *Nat Protoc* **13**, 1006-1019, doi:10.1038/nprot.2018.015 (2018).
- 13 Meers, M. P., Bryson, T. D., Henikoff, J. G. & Henikoff, S. Improved CUT&RUN chromatin profiling tools. *Elife* **8**, doi:10.7554/eLife.46314 (2019).
- 14 Proffitt, K. D. & Virshup, D. M. Precise regulation of porcupine activity is required for physiological Wnt signaling. *J Biol Chem* **287**, 34167-34178, doi:10.1074/jbc.M112.381970 (2012).
- 15 Chen, B. *et al.* Small molecule-mediated disruption of Wnt-dependent signaling in tissue regeneration and cancer. *Nat Chem Biol* **5**, 100-107, doi:10.1038/nchembio.137 (2009).
- 16 Lewis, S. L. *et al.* Dkk1 and Wnt3 interact to control head morphogenesis in the mouse. *Development* **135**, 1791-1801, doi:10.1242/dev.018853 (2008).
- 17 Semenov, M. V., Zhang, X. & He, X. DKK1 antagonizes Wnt signaling without promotion of LRP6 internalization and degradation. *J Biol Chem* **283**, 21427-21432, doi:10.1074/jbc.M800014200 (2008).
- 18 Wu, H. *et al.* Histone methyltransferase G9a contributes to H3K27 methylation in vivo. *Cell Res* **21**, 365-367, doi:10.1038/cr.2010.157 (2011).
- 19 Tachibana, M., Sugimoto, K., Fukushima, T. & Shinkai, Y. Set domain-containing protein, G9a, is a novel lysine-preferring mammalian histone methyltransferase with hyperactivity and specific selectivity to lysines 9 and 27 of histone H3. *J Biol Chem* **276**, 25309-25317, doi:10.1074/jbc.M101914200 (2001).
- 20 Patnaik, D. *et al.* Substrate specificity and kinetic mechanism of mammalian G9a histone H3 methyltransferase. *J Biol Chem* **279**, 53248-53258, doi:10.1074/jbc.M409604200 (2004).

- 21 Mozzetta, C. *et al.* The histone H3 lysine 9 methyltransferases G9a and GLP regulate polycomb repressive complex 2-mediated gene silencing. *Mol Cell* **53**, 277-289, doi:10.1016/j.molcel.2013.12.005 (2014).
- 22 Mochizuki-Kashio, M. *et al.* Dependency on the polycomb gene Ezh2 distinguishes fetal from adult hematopoietic stem cells. *Blood* **118**, 6553-6561, doi:10.1182/blood-2011-03-340554 (2011).
- 23 Xie, H. *et al.* Polycomb repressive complex 2 regulates normal hematopoietic stem cell function in a developmental-stage-specific manner. *Cell Stem Cell* **14**, 68-80, doi:10.1016/j.stem.2013.10.001 (2014).

Reviewers' Comments:

Reviewer #1:

Remarks to the Author:

The authors have addressed my concern and additional data strengthen the manuscript. This paper has revealed the unappreciated mechanisms through which Ezh2 is involved in functional EMP production in the mouse embryo.

Reviewer #2:

Remarks to the Author:

This referee appreciates authors' great effort to address all questions raised by referees. Most of my concerns have been addressed, and I find the revised manuscript much improved and suitable for publication.

Reviewer #3:

Remarks to the Author:

I have no further concerns and support publication of this well-conducted study.